# MULTI-LABEL TEST-TIME ADAPTATION WITH BOUND ENTROPY MINIMIZATION

**Xiangyu Wu**[1,2]**, Feng Yu**[1]**, Qing-Guo Chen**[2]**, Yang Yang**[1]*, **Jianfeng Lu**[1]*
[1]Nanjing University of Science and Technology
[2]Alibaba International Digital Commerce Group

## ABSTRACT

Mainstream test-time adaptation (TTA) techniques endeavor to mitigate distribution shifts via entropy minimization for multi-class classification, inherently increasing the probability of the most confident class. However, when encountering multi-label instances, the primary challenge stems from the varying number of labels per image, and prioritizing only the highest probability class inevitably undermines the adaptation of other positive labels. To address this issue, we investigate TTA within multi-label scenario (**ML–TTA**), developing **B**ound **E**ntropy **M**inimization (**BEM**) objective to simultaneously increase the confidence of multiple *top* predicted labels. Specifically, to determine the number of labels for each augmented view, we retrieve a paired caption with yielded textual labels for that view. These labels are allocated to both the view and caption, called *weak label set* and *strong label set* with the same size $k$. Following this, the proposed BEM considers the highest *top-k* predicted labels from view and caption as a single entity, respectively, learning both view and caption prompts concurrently. By binding *top-k* predicted labels, BEM overcomes the limitation of vanilla entropy minimization, which exclusively optimizes the most confident class. Across the MSCOCO, VOC, and NUSWIDE multi-label datasets, our ML–TTA framework equipped with BEM exhibits superior performance compared to the latest SOTA methods, across various model architectures, prompt initialization, and varying label scenarios. The code is available at https://github.com/Jinx630/ML-TTA.

## 1 INTRODUCTION

The advent of vision-language models (VLMs) (Radford et al., 2021; Li et al., 2023; Zeng et al., 2024; Yang et al., 2024a) has facilitated remarkable generalization capabilities by pretraining on massive datasets. Nonetheless, VLMs such as CLIP (Radford et al., 2021), require sophisticated prompt learning when confronted with considerable discrepancies between training and testing domains, to prevent performance degradation due to distribution shifts occurring during testing time.

Fortunately, recent advancements (Shu et al., 2022; Feng et al., 2023; Ma et al., 2023; Liu et al., 2024b; Zhang et al., 2024b; Zhao et al., 2024a; Karmanov et al., 2024; Yoon et al., 2024; Gao et al., 2024) allow for immediate adaptation to any distribution of test instance during testing time, which is known as Test-Time Adaptation (TTA). As pioneering works, TPT (Shu et al., 2022) and its enhancement, DiffTPT (Feng et al., 2023), select a set of confident augmented views, learning instance-level prompt for each test instance. DART (Liu et al., 2024b) and DMN (Zhang et al., 2024b), to fully utilize the encountered knowledge from past samples, design dual-modal knowledge retention prompts and dynamic dual-memory networks, respectively, to adaptively incorporate historical knowledge. The central premise of these methods is entropy minimization, which aims to minimize inconsistency and uncertainty over the model predictions, and further increase the prediction probability of the highest confidence class, a theory that is readily demonstrable.

Although entropy loss is advantageous for TTA as an uncertainty metric, a natural question arises: Can it be reliably applied to instances with multiple positive labels? As illustrated in Figure 1 (a), for the positive label set {*keyboard, phone, remote, mouse, book*}, compared to CLIP, all methods consistently boost the probability of the most confident class, *keyboard*. Nonetheless, TPT (Shu

---

*Corresponding author

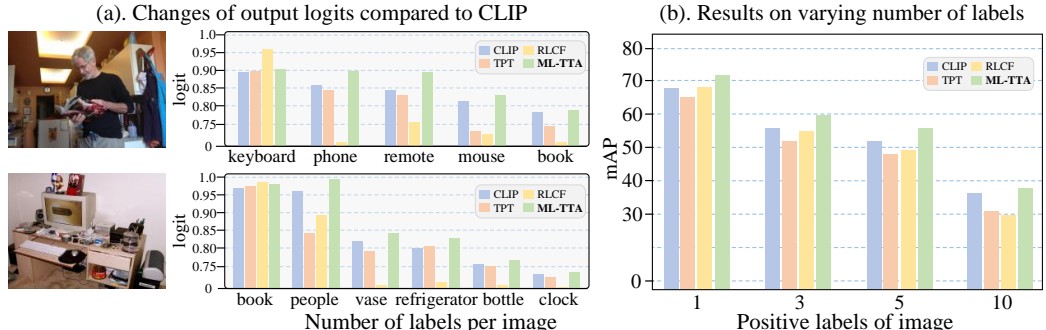

Figure 1: (a). Compared to CLIP (Radford et al., 2021), ML–TTA increases all positive label logits simultaneously, while others focus only on *top-1* class. (b). Comparison of various methods on images with varying numbers. Compared to CLIP, as the number of labels per image rises, the adaptability of TPT (Shu et al., 2022) and RLCF (Zhao et al., 2024a) in handling multi-label images shows a marked decrease.

et al., 2022) and RLCF (Zhao et al., 2024a) adversely impair the remaining positive labels. This indicates that existing TTA methods primarily focus on increasing the confidence of *top-1* label, leading to insufficient adaptation for other positive labels. Given this, we expect to treat the highest *top-k* positive labels as a single label, aiming to simultaneously increase the predicted confidence of multiple *top-k* labels. However, positive label sets are not known in advance in real applications.

Based on the preceding discussion, we investigate the TTA within multi-label scenario (**ML–TTA**) and propose a novel theoretical optimization objective named **B**ound **E**ntropy **M**inimization (**BEM**), which posits that when the highest *top-k* predicted labels (*k* being the size of positive label set) share identical probabilities, the entropy loss will uniformly increase the probabilities of all *top-k* classes. Consider a multi-label test image with a set of augmented views, to determine the number of positive labels for each view, we retrieve a paired caption with derived textual labels for each view, which then serves as *weak label set* of size *k* for the corresponding view. Furthermore, owing to the aligned visual-language space of CLIP (Radford et al., 2021), texts can be treated as pseudo-images with known positive labels, a premise corroborated by recent academic research (Guo et al., 2023; Zhao et al., 2024b; Li et al., 2024a; Wu et al., 2024). Drawing inspiration from these findings, we conceptualize each paired caption as a pseudo-view possessing a known label set, termed *strong label set*, of the same size *k*, since the textual labels are directly derived from captions.

Upon determining the *weak label set* for each view and the *strong label set* for each paired caption, the proposed BEM objective binds the highest *top-k* predicted labels as a single label for both view and caption. By optimizing the view prompt and caption prompt, the model is encouraged to concurrently increase the confidence of the *top-k* classes. Additionally, since some augmented views and paired captions may fail to capture the target label area, leading to misleading predictions, we adopt *confidence selection* utilized in TPT (Shu et al., 2022) to filter out "noisy" views and captions with high entropy (*i.e.*, low confidence). Consequently, in this paper, starting from TPT, the developed ML–TTA framework equipped with BEM endows the CLIP's adaptability of multi-label instances during testing. Our contributions are summarized as follows:

- We examine the **M**ulti-**L**abel **T**est-**T**ime **A**daptation (ML–TTA) and propose **B**ound **E**ntropy **M**inimization (BEM), which simultaneously increase the probabilities of all highest *top* labels.
- BEM binds *weak label set* of view and *strong label set* of the caption as a single label, respectively, learning instance-level view and caption prompts for adapting multi-label test instances.
- On the MSCOCO, VOC, and NUSWIDE datasets, ML–TTA outperforms the original CLIP model as well as other state-of-the-art TTA methods designed for multi-class classification, across various model architectures, prompt initialization, and varying label scenarios.

## 2 RELATED WORK

### 2.1 TEST-TIME ADAPTION

Test-time adaptation (TTA) (Zhang et al., 2022; Shu et al., 2022; Ma et al., 2023; Karmanov et al., 2024; Zhao et al., 2024a; Lee et al., 2024; Chi et al., 2024; Ma et al., 2024) enables models to adapt

changing distributions during testing time without accessing to the source domain data or extensive target domain data. Within the spectrum of TTA settings, *e.g.*, "fully" TTA (Wang et al., 2021; Zhao et al., 2023), "online" TTA (Lee & Chang, 2024; Lee et al., 2024), "continuous" TTA (Liu et al., 2024a; Song et al., 2023), and "prior" TTA (Wei et al., 2023; 2024), "online" TTA (Shu et al., 2022; Karmanov et al., 2024; Zhao et al., 2024a) focuses on adapting to individual samples and is particularly valuable in many application domains, such as autonomous driving, where weather conditions are constantly changing, and road monitoring, where traffic patterns are continually evolving. MEMO (Zhang et al., 2022) is the pioneering work that proposes consistent predictions across diverse augmented views. Following this, TPT (Shu et al., 2022) notably enhances the generalization capabilities of the CLIP (Radford et al., 2021) model to unseen test data by entropy minimization. SwapPrompt (Ma et al., 2023) utilizes online and target prompts, enhancing the CLIP's adaptability by preserving historical information and alternating prediction. In contrast, TDA (Karmanov et al., 2024) adapts to streaming input data by constructing a dynamic key-value cache from historical data. RLCF (Zhao et al., 2024a) incorporates reinforcement learning to distill knowledge into more compact models. Among these works, MEMO (Zhang et al., 2022), TPT (Shu et al., 2022), and RLCF (Zhao et al., 2024a) are particularly challenging, as the model is reset after adapting a test instance, obviating the need to retain historical knowledge, and thereby accommodating continuously shifting test distributions. Nonetheless, these methods are primarily designed for multi-class classification and may not be as effective in the more common multi-label scenario.

## 2.2 PROMPT LEARNING IN VLMS

Visual-language models (VLMs) (Li et al., 2021; Wu et al., 2022; Yang et al., 2024b; Li et al., 2023; Wan et al., 2024; Zeng et al., 2024; Huang et al., 2024), trained on massive image-text pairs (Sharma et al., 2018; Schuhmann et al., 2022), have demonstrated remarkable proficiency in cross-task learning. To further enhance the transfer abilities of CLIP (Radford et al., 2021), researchers have developed various prompt learning techniques (Zhou et al., 2022b;a; Fu et al., 2024; Li et al., 2024b; Wu et al., 2024). For instance, the groundbreaking work CoOp (Zhou et al., 2022b), and its advancement CoCoOp (Zhou et al., 2022a), are the first to propose optimizing context vectors to improve the generalization capabilities of CLIP. Maple (Khattak et al., 2023) introduces a multimodal prompt learning method, designed to recalibrate both visual and language modalities. Dept (Zhang et al., 2024a) and PromptKD (Li et al., 2024b) take on the challenge from the perspectives of knowledge retention and distillation, respectively, to promote robust generalization on novel tasks. Exploiting the aligned visual-language space of CLIP (Radford et al., 2021), TAI-DPT (Guo et al., 2023), PVP (Wu et al., 2024) and RC-TPL (Zhao et al., 2024b) propose to regard texts as images for prompt tuning in zero-shot multi-label image classification. Investigations like DualCoOp (Sun et al., 2022), DualCoOp++ (Hu et al., 2023), and VLPL (Xing et al., 2024) consider more intricate tasks, enhancing multi-label classification capabilities in the partial-label scenario. In contrast, our study focuses on a training-free paradigm, termed multi-label test-time adaptation, which obviates the need for the source training data and is exclusively at the testing instance level.

## 3 METHOD

In Sec. 3.1, we review the entropy minimization widely used in TTA. In Sec. 3.2, we highlight the issue that vanilla entropy minimization predominantly increases the probability of *top-1* predicted label and propose a new proposition **B**ound **E**ntropy **M**inimization (BEM). In Sec. 3.3, we present a **M**ulti-**L**abel **T**est-**T**ime **A**daptation (ML–TTA) framework, incorporating BEM, which binds the highest *top* predicted labels of both augmented views and paired captions as an individual single label. ML–TTA consists of view-caption constructing (Sec. 3.3.1) and label binding (Sec. 3.3.2).

## 3.1 PRELIMINARIES

The purpose of Test-Time Adaptation is to utilize each test instance once for immediate adaptation before inference, without any prior assumptions about the test data distribution. For the TTA of VLMs, let $\mathcal{M}_\theta$ denote the CLIP model trained on the training dataset $\mathcal{D}^{\text{train}} = \{(\mathbf{x}_i^{\text{train}}, \mathbf{y}_i^{\text{train}}) \mid \mathbf{x}_i^{\text{train}} \in \mathcal{X}^{\text{train}}, \mathbf{y}_i^{\text{train}} \in \mathcal{Y}^{\text{train}}\}_{i=1}^{M^{\text{train}}}$. The TTA approach, TPT (Shu et al., 2022), incorporates the Marginal Entropy Minimization (MEM) objective to adapt $\mathcal{M}_\theta$ using a solitary instance $\mathbf{x}^{\text{test}}$ from the testing dataset $\mathcal{D}^{\text{test}} = \{(\mathbf{x}_i^{\text{test}}, \mathbf{y}_i^{\text{test}}) \mid \mathbf{x}_i^{\text{test}} \in \mathcal{X}^{\text{test}}, \mathbf{y}_i^{\text{test}} \in \mathcal{Y}^{\text{test}}\}_{i=1}^{M^{\text{test}}}$.

Given a test instance $\mathbf{x}^{\text{test}}$ and a set $\mathcal{A}$ of $N$ random augmentation functions, $\mathbf{x}^{\text{test}}$ is first augmented $N$ times to generate a set of different views, represented as $\mathbf{X}^{\text{test}} = \{\mathbf{x}_j^{\text{test}} \mid \mathbf{x}_j^{\text{test}} = \mathcal{A}_j(\mathbf{x}^{\text{test}})\}_{j=1}^N$. TTA aims to minimize the marginal entropy of these augmented views, encouraging the model to perform consistent and confident predictions. The entropy of an augmented view is defined as:

$$H(p(\cdot|\mathbf{x}_j^{\text{test}})) = -\sum_{l=1}^{L} p(y = l|\mathbf{x}_j^{\text{test}}) \log(p(y = l|\mathbf{x}_j^{\text{test}})), \tag{1}$$

where $l \in \mathcal{Y}^{\text{test}}$ and $L$ is the number of labels in $\mathcal{Y}^{\text{test}}$. The core principle of TPT is to minimize the marginal entropy of the prediction probability distributions of selected confident augmented views by a ratio $\tau$, thereby encouraging the model to make consistent predictions. After obtaining the average entropy of these confident views, denoted as $\tilde{H}$, TPT updates the prompt using a single gradient descent step based on $\tilde{H}$ and performs immediate inference on this test instance. Once inference is done, the model's prompt and optimizer are reset promptly for adaptation to the next test instance. Owing to its simplicity and effectiveness, Marginal Entropy Minimization has emerged as a *de facto* standard in modern TTA.

### 3.2 BOUND ENTROPY MINIMIZATION

It can be observed that the TPT method selects a subset of confident augmented views with lower entropy (*i.e.*, high confidence) from $\mathbf{X}^{\text{test}}$, continually minimizing the average entropy of these confident views to maintain consistent model predictions across these views. With respect to vanilla entropy minimization within TTA, the following proposition holds.

**Proposition 1.** *Consider the output logits of a confident view $x$, denoted as $\mathbf{s} = (s_1, s_2, \ldots, s_L)$, where, without loss of generality, we assume $s_1 > s_2 > \cdots > s_L$. It can be deduced that the entropy loss $H = H(p(\cdot|x))$ decreases as $s_1$ increases, and $H$ increases as the sum of the remaining logits, $S_{rest} = \sum_{i=2}^{L} s_i$, decreases. Formally, this relationship can be expressed as:*

$$\nabla_{s_1} H = \frac{\partial H}{\partial s_1} < 0 \quad and \quad \nabla_{s_{rest}} H = \frac{\partial H}{\partial S_{rest}} > 0. \tag{2}$$

A detailed proof is provided in the Appendix. Following a single gradient descent update step, we can derive $s_1^{(t+1)} = s_1^{(t)} - \alpha \nabla_{s_1} H$ and $S_{rest}^{(t+1)} = S_{rest}^{(t)} - \alpha \nabla_{S_{rest}} H$, where $\alpha$ denotes the learning rate. Therefore, Proposition 1 indicates that the nature of entropy loss is to increase the probability of the most confident class while diminishing the cumulative probability of the rest classes. Hence, when adapting to single-label test instances, the goal of vanilla entropy minimization is to solely maximize the probability of the *top-1* predicted label, disregarding changes in the probabilities of the remaining labels.

In contrast, in the context of multi-label test-time adaptation, where the test instance may include a set of positive labels $L_p = \{l_{p1}, l_{p2}, ..., l_{pk}\}$. In this case, regardless of whether the *top-1* predicted label is the element of the positive label set $L_p$, the entropy loss will inevitably decrease the prediction probabilities of the other positive labels within $L_p$ while increasing the probability of the most confident class. This may lead to the model overemphasizing the *top-1* predicted label and inadequately adapting to the other positive labels. Therefore, for test-time adaptation in multi-label data, we propose the following proposition, termed Bound Entropy Minimization.

**Proposition 2.** *Consider the output logits of a confident view $x$, denoted as $\mathbf{s} = (s_1, s_2, \ldots, s_L)$, where, without loss of generality, we assume $s_1 > s_2 > \cdots > s_L$. We define the modified logits as $\mathbf{s}' = (s_1', s_2', \ldots, s_L')$, where $s_i' = a_i + s_i$ for $i \leq k$ with $a_i = s_1 - s_i$ and $s_i' = s_i$ for $i > k$. Here, $a_i$ is a constant value that does not participate in differentiation, resulting in $s_i' = s_1$ for all $i \leq k$. Let $S_{rest} = \sum_{i=k+1}^{L} s_i$. For the modified logits $\mathbf{s}'$, we define the modified probability $\mathbf{p}' = Softmax(\mathbf{s}')$, and the modified entropy as $H' = -\sum_{i=1}^{L} p_i' \log p_i'$. It follows that:*

$$\nabla_{s_i} H' = \frac{\partial H'}{\partial s_i} < 0, \quad \forall i \leq k \quad and \quad \nabla_{s_{rest}} H' = \frac{\partial H'}{\partial S_{rest}} > 0. \tag{3}$$

A detailed proof is provided in Appendix. Likewise, after one step of gradient descent optimization, the prediction probabilities of all *top-k* predicted labels will further increase due to $\nabla_{s_i} H' < 0$ for all $i <= k$ and $\nabla_{s_{rest}} H' > 0$. Therefore, from Proposition 2, to be robust against distribution shifts with multiple labels, it is crucial to determine the number of positive labels for adapting multi-label test instances. In the following subsection, we will introduce a novel Multi-Label Test-Time Adaptation framework by employing proposition 2 and incorporating text captions into the adaptation system.

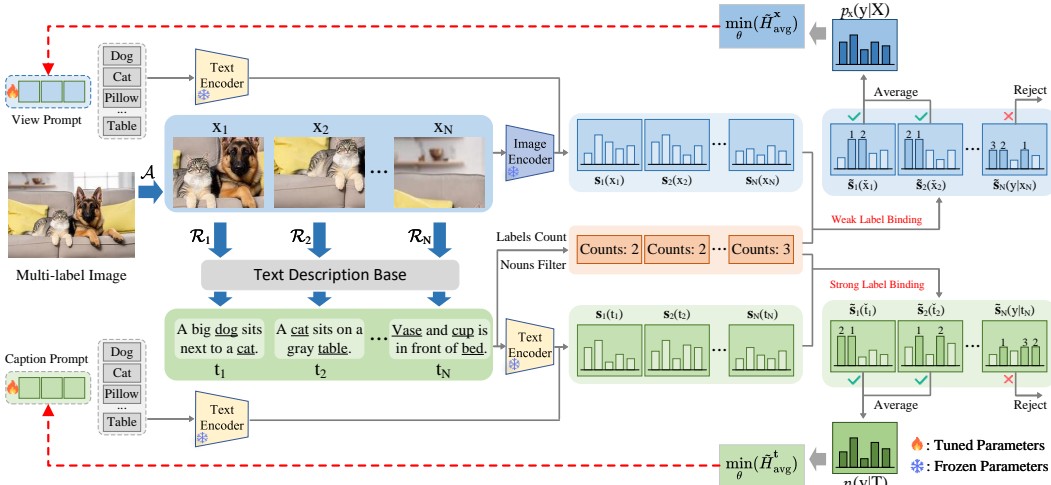

Figure 2: Overview of proposed multi-label test-time adaption.

## 3.3 MULTI-LABEL TEST-TIME ADAPTATION

### 3.3.1 VIEW-CAPTION CONSTRUCTING

Benefiting from the aligned space of CLIP, any image can be assigned a most similar caption from an offline text description base based on similarity retrieval. As depicted in Figure 2, given a test image $\mathbf{x}^{\text{test}}$ and a collection of random augmentation functions $\mathcal{A} = \{\mathcal{A}_1, \mathcal{A}_2, ..., \mathcal{A}_N\}$, $\mathbf{x}^{\text{test}}$ is first augmented $N$ times to generate a set of different augmented views. For each augmented view, we retrieve the most similar caption from an offline text description database to serve as its paired caption. The views generating and caption allocating can be expressed as:

$$X^{\text{test}} = \{\mathbf{x}_i^{\text{test}} \mid \mathbf{x}_i^{\text{test}} = \mathcal{A}_i(\mathbf{x}^{\text{test}})\}_{i=1}^{N}, \ T^{\text{test}} = \{\mathbf{t}_i^{\text{test}} \mid \mathbf{t}_i^{\text{test}} = \mathcal{R}_i(\mathbf{x}_i^{\text{test}})\}_{i=1}^{N}, \tag{4}$$

where $\mathcal{A}_i$ and $\mathcal{R}_i$ represents augmentation and retrieval by computing similarity. To streamline the retrieval process, we directly utilize the method proposed in PVP (Wu et al., 2024), which employs LLama-2-7B (Touvron et al., 2023) to construct the text description base, each text is a description of a natural scene containing several categories. Then, CLIP is used to extract text embeddings and construct an offline database of size $B \times d$, where $B$ denotes the number of test descriptions and $d$ denotes the embedding dimension. More details of the text description base construction are provided in the appendix.

The goal of TTA is to calibrate the model for a single unlabeled test instance. Clearly, a single instance is insufficient for tuning the entire CLIP model to learn domain-specific knowledge. Consequently, as shown in Figure 2, akin to prompt tuning paradigm, we design two identical prompts, referred to as view prompt and caption prompt, denoted by $\mathbf{V}$ and $\mathbf{C}$, respectively. Treating prompt tuning at test-time as a way to furnish customized context for individual test instances. Benefiting from the aligned space of CLIP, the representations of images and texts share similar semantic information, therefore, the paired caption can be considered as a "pseudo image" with accurate textual labels, encouraging the model to learn visual-related knowledge and complementary information from views and captions jointly. For $L$ categories, we initialize the view and caption prompts with template "*a photo of a* $[\mathbf{CLS}]_j$", in which $[\mathbf{CLS}]_j$ represents the $j$-th label name, *e.g.*, dog or cat, results in $\mathbf{v}_j$ and $\mathbf{c}_j$. Once the paired views and captions are obtained, we compute the logits for each view $\mathbf{x}_i^{\text{test}}$ on $L$ view prompts and for each caption $\mathbf{t}_i^{\text{test}}$ on $L$ caption prompts as below:

$$s_{ij}^{\mathbf{x}_i^{\text{test}}} = \langle \text{Enc}^{\text{I}}(\mathbf{x}_i^{\text{test}}), \text{Enc}^{\text{T}}(\mathbf{v}_j) \rangle, s_{ij}^{\mathbf{t}_i^{\text{test}}} = \langle \text{Enc}^{\text{T}}(\mathbf{t}_i^{\text{test}}), \text{Enc}^{\text{T}}(\mathbf{c}_j) \rangle, \tag{5}$$

where $\text{Enc}^{\text{I}}$ and $\text{Enc}^{\text{T}}$ represent the frozen image encoder and text encoder of CLIP, $\langle \cdot, \cdot \rangle$ signifies the dot product. As stated in proposition 2, the crux of adapting multi-label instance lies in identifying the size of positive label set for each view $\mathbf{x}_i^{\text{test}}$ and caption $\mathbf{t}_i^{\text{test}}$.

---

**Algorithm 1:** Label Binding Algorithm

---

**Input:** Logits $\mathbf{s}_i$ before label binding and the size of weak label set $k^{\mathbf{x}_i}$.

**Output:** Modified logits $\tilde{\mathbf{s}}_i$ after label binding.

1   $m_i = \max_j s_{ij}$ ;
2   **for** $j = 1$ **to** $L$ **do**
3      $a_{ij} = \text{detach}\,(m_i - s_{ij})$               ▷ Detach from gradient. ;
4      **if** $\text{Rank}_{(s_{ij}, \mathbf{s}_i)} \leq k^{\mathbf{x}_i}$ **then**
5         $\tilde{s}_{ij} = a_{ij} + s_{ij}$     ▷ Bind $s_{ij}$ if $j$-th label is in highest *top-$k^{\mathbf{x}_i}$* predicted labels. ;
6      **end if**
7      **else**
8         $\tilde{s}_{ij} = s_{ij}$ ;
9      **end if**
10   **end for**
11   $\tilde{\mathbf{s}}_i = (\tilde{s}_{i0}, \tilde{s}_{i1}, \cdots, \tilde{s}_{iL})$

---

### 3.3.2   LABEL BINDING

Obviously, the positive label set for $\mathbf{x}_i^{\text{test}}$ is not feasible to obtain directly. Fortunately, the textual labels for $\mathbf{t}_i^{\text{test}}$, which we refer to as *strong label set*, can be readily derived through noun filtering, *e.g.*, *A truck drives past a black car with a suitcase on top.* with extracted *strong label set* being *truck, car, suitcase*. Moreover, this set can also serve as a pseudo-positive label set, termed the *weak label set*, for $\mathbf{x}_i^{\text{test}}$. Consequently, we treat the size of *strong label set* as the *top-k* bound highest logits of captions, akin to views. The binding operation for $s_{ij}^{\mathbf{x}_i^{\text{test}}}$ and $s_{ij}^{\mathbf{t}_i^{\text{test}}}$ can be expressed as:

$$
\begin{aligned}
\tilde{s}_{ij}^{\mathbf{x}^{\text{test}}} &= ((m_i^{\mathbf{x}^{\text{test}}} - s_{ij}^{\mathbf{x}^{\text{test}}}) + s_{ij}^{\mathbf{x}^{\text{test}}}) \cdot \mathbb{I}(\text{Rank}_{(s_{ij}^{\mathbf{x}^{\text{test}}}, \mathbf{s}_i^{\mathbf{x}^{\text{test}}})} \leq k^{\mathbf{x}_i^{\text{test}}}) + s_{ij}^{\mathbf{x}^{\text{test}}} \cdot \mathbb{I}(\text{Rank}_{(s_{ij}^{\mathbf{x}^{\text{test}}}, \mathbf{s}_i^{\mathbf{x}^{\text{test}}})} > k^{\mathbf{x}_i^{\text{test}}}), \\
\tilde{s}_{ij}^{\mathbf{t}^{\text{test}}} &= ((m_i^{\mathbf{t}^{\text{test}}} - s_{ij}^{\mathbf{t}^{\text{test}}}) + s_{ij}^{\mathbf{t}^{\text{test}}}) \cdot \mathbb{I}(\text{Rank}_{(s_{ij}^{\mathbf{t}^{\text{test}}}, \mathbf{s}_i^{\mathbf{t}^{\text{test}}})} \leq k^{\mathbf{t}_i^{\text{test}}}) + s_{ij}^{\mathbf{t}^{\text{test}}} \cdot \mathbb{I}(\text{Rank}_{(s_{ij}^{\mathbf{t}^{\text{test}}}, \mathbf{s}_i^{\mathbf{t}^{\text{test}}})} > k^{\mathbf{t}_i^{\text{test}}}),
\end{aligned}
\tag{6}
$$

where $((m_i^{\mathbf{x}^{\text{test}}} - s_{ij}^{\mathbf{x}^{\text{test}}}) + s_{ij}^{\mathbf{x}^{\text{test}}})$ employs stop-gradient operation follow VQ-VAE van den Oord et al. (2017), $\mathbf{s}_i^{\mathbf{x}^{\text{test}}} = (s_{i1}^{\mathbf{x}^{\text{test}}}, s_{i2}^{\mathbf{x}^{\text{test}}}, ..., s_{iL}^{\mathbf{x}^{\text{test}}})$ and $\mathbf{s}_i^{\mathbf{t}^{\text{test}}} = (s_{i1}^{\mathbf{t}^{\text{test}}}, s_{i2}^{\mathbf{t}^{\text{test}}}, ..., s_{iL}^{\mathbf{t}^{\text{test}}})$ denotes the logits before binding, $m_i^{\mathbf{x}^{\text{test}}}$ and $m_i^{\mathbf{t}^{\text{test}}}$ denotes the maximum logit of $\mathbf{s}_i^{\mathbf{x}^{\text{test}}}$ and $\mathbf{s}_i^{\mathbf{t}^{\text{test}}}$, respectively, $\mathbb{I}(\cdot)$ denotes the indicator function, and $\text{Rank}(s, \mathbf{s})$ indicates the descending rank of $s$ within $\mathbf{s}$, $k^{\mathbf{x}_i^{\text{test}}}$ and $k^{\mathbf{t}_i^{\text{test}}}$ denotes the size of *weak label set* of $i$-th augmented view and *strong label set* of $i$-th paired caption. The algorithm process of label binding is presented in algorithm 1. We provide a detailed label binding process using a 3-class classification task in the Appendix.

To reduce the noise brought by random augmentation and the noise in the caption caused by noisy views, we employ *confidence selection* to filter out noisy views and captions with higher entropy (*i.e.*, lower confidence). Such noisy views may, due to random cropping augmentation, exclude the target label area, leaving only irrelevant background information. Similarly, the retrieved paired captions for these noisy views will lack any pertinent textual labels. We selected views and captions with lower predicted entropy by a ratio $\tau$, yielding $\{\check{\mathbf{x}}_i^{\text{test}}\}_{i=1}^{\tau N}$ for views and $\{\check{\mathbf{t}}_i^{\text{test}}\}_{i=1}^{\tau N}$ for captions.

Taking views $\check{\mathbf{x}}_i^{\text{test}}$ as an example, the probability of $\check{\mathbf{x}}_i^{\text{test}}$ on $L$ labels denoted as $\mathbf{p} = \text{Softmax}(\tilde{\mathbf{s}}_i^{\check{\mathbf{x}}_i^{\text{test}}})$, the average predicted entropy of the filtered low-entropy views can be expressed as:

$$
\tilde{H}_{\text{avg}}^{\check{\mathbf{x}}^{\text{test}}} = \frac{1}{\tau N} \sum_{i=1}^{\tau N} \left( - \sum_{l=1}^{L} p(y = l | \check{\mathbf{x}}_i^{\text{test}}) \log(p(y = l | \check{\mathbf{x}}_i^{\text{test}})) \right).
\tag{7}
$$

Subsequently, the bound entropy optimization objective of view prompt $\mathbf{V}$ is to minimize the predicted entropy through $\tilde{H}_{\text{avg}}^{\check{\mathbf{x}}^{\text{test}}}$. For the objective of caption prompt $\mathbf{C}$, we replace $\check{\mathbf{x}}_i^{\text{test}}$ in Eq.(7) with confident captions $\check{\mathbf{t}}_i^{\text{test}}$ to obtain $\tilde{H}_{\text{avg}}^{\check{\mathbf{t}}^{\text{test}}}$.

### 3.3.3   OVERALL OBJECTIVE OF ML–TTA

ML–TTA calculates the predicted bound entropy of confident augmented views and paired captions, optimizing both view prompt and caption prompt with a single step of gradient descent, and simultaneously increasing the probability of highest *top* predicted labels. Then, the overall bound entropy

loss is given by:

$$\tilde{H}_{\text{BEM}} = \tilde{H}_{\text{avg}}^{\check{\mathbf{x}}^{\text{test}}} + \tilde{H}_{\text{avg}}^{\check{\boldsymbol{t}}^{\text{test}}}. \tag{8}$$

After optimizing the prompts, ML–TTA immediately infers the test instance $\mathbf{x}^{\text{test}}$ and resets the parameters of the prompts ($\mathbf{V}$ and $\mathbf{C}$) and the state of optimizer to adapt to the next test instance. During the inference phase, we separately compute the similarity between the view prompt $\mathbf{V}$ and the test instance $\mathbf{x}^{\text{test}}$, as well as the similarity between the caption prompt $\mathbf{C}$ and the test instance $\mathbf{x}^{\text{test}}$, and directly add these two similarities to obtain the final prediction result.

## 4  EXPERIMENT

### 4.1  EXPERIMENTAL SETUP

**Benchmarks.** We utilize the widely employed CLIP (Radford et al., 2021) model as source model and select the multi-label datasets VOC (Everingham et al., 2010), MSCOCO (Lin et al., 2014), and NUSWIDE (Chua et al., 2009) as target domains. The VOC dataset includes 20 categories, covering both VOC2007 and VOC2012 versions, which contain 4,952 and 5,823 test images, respectively. The MSCOCO dataset extends the category range to 80, and for testing purposes, we employ the validation sets of COCO2014 with 40,504 images and COCO2017 with 5,000 images, as the test set labels are not accessible. The NUSWIDE dataset includes 81 categories with a total of 83,898 test images of lower resolution, which presents a broader category spectrum than MSCOCO.

**Implementation details.** All experiments are based on the CLIP model, encompassing RN50, RN101, ViT-B/32, and ViT-B/16 architectures, each consisting of an image encoder and a corresponding text encoder. For the initialization of the view and caption prompts, we employ the token embedding of the "a photo of a" hard prompt as initialization weights and another using learned prompts from CoOp (Zhou et al., 2022b) and MaPLE (Khattak et al., 2023). The learning rate for the view prompt is 1e-2, while for the caption prompt is 1e-3. For all settings, multi-label test-time adaptation is performed on a single instance, *i.e.*, the batch size is 1. The ratio for filtering confident views and captions is 0.1. The optimizer is AdamW (Loshchilov & Hutter, 2019) with a single update step, followed by immediate inference on the test instance. Following PVP (Wu et al., 2024), we collect 100k text descriptions for each dataset, resulting in a total size of 300k text description base. All experiments are evaluated by the mean Average Precision (mAP) metric, defined as $mAP = \frac{1}{L}\sum_{i=1}^{L} AP_i$, where $L$ is the number of categories, and $AP_i$ is the area under the Precision-Recall curve for the $i$-th category.

### 4.2  COMPARISONS WITH STATE-OF-THE-ART

To our knowledge, our work is the first to investigate the feasibility of traditional entropy minimization in the multi-label setting. Therefore, in this section, we select the original CLIP model and other SOTA methods for multi-class scenarios as baselines, including *episdoic* methods that do not require retaining historical knowledge (TPT Shu et al. (2022), DiffTPT Feng et al. (2023), RCLF Zhao et al. (2024a)) and *online* methods that do (DMN Zhang et al. (2024b), TDA Karmanov et al. (2024)).

**Results on different architectures.** Table 1 compares ML–TTA with both *online* and *episdoic* TTA methods on different CLIP (Radford et al., 2021) architectures, demonstrating the superior performance across various multi-label datasets. Specifically, for the RN50 and RN101 architectures on COCO2014/2017 (Lin et al., 2014) datasets, ML–TTA achieves 4∼5% improvement in mAP over the original CLIP (Radford et al., 2021) model, whereas TPT (Shu et al., 2022) and DiffTPT (Feng et al., 2023) yield only 1% enhancement. Despite introducing dual-memory network knowledge from historical samples, DMN (Zhang et al., 2024b) and TDA Karmanov et al. (2024) present a slight performance decline, due to intensifying the optimization bias towards *top-1* label. Notably, RLCF (Zhao et al., 2024a) employs a reinforcement learning-based knowledge distillation and more adaptation steps, resulting in a catastrophic degradation in the multi-label adaptation performance for smaller models due to excessive optimizations of *top-1* label. On the VOC2012/2017 (Everingham et al., 2010) datasets, TPT and DiffTPT also show 1∼2% decrease in performance compared to CLIP, whereas ML–TTA still maintains 2∼3% performance improvement, indicating the robustness of ML–TTA in multi-label adaptation across various model architectures and datasets.

Table 1: Comparison with CLIP and SOTAs on adapting multi-label instances with different architectures.

| | Methods | *Epsdoic* | COCO2014 | COCO2017 | VOC2007 | VOC2012 | NUSWIDE | Average |
|---|---|---|---|---|---|---|---|---|
| **RN-50** | CLIP [ICML 2022] | ✓ | 47.53 | 47.32 | 75.91 | 74.25 | 41.53 | 57.31 |
| | DMN [CVPR 2024] | ✗ | 44.54 | 44.18 | 74.87 | 74.13 | 41.32 | 55.81 |
| | TDA [CVPR 2024] | ✗ | 48.91 | 49.11 | 76.64 | 75.12 | 42.34 | 58.42 |
| | TPT [NeurIPS 2022] | ✓ | 48.52 | 48.51 | 75.54 | 73.92 | 41.97 | 57.69 |
| | DIffTPT [ICCV 2023] | ✓ | 48.56 | 48.67 | 75.89 | 74.13 | 41.33 | 57.72 |
| | RLCF [ICLR 2024] | ✓ | 36.87 | 36.73 | 65.75 | 64.73 | 29.83 | 46.78 |
| | **ML–TTA (Ours)** | ✓ | **51.58** | **51.39** | **78.62** | **76.63** | **42.53** | **60.15** |
| **RN-101** | CLIP [ICML 2022] | ✓ | 48.83 | 48.15 | 76.72 | 74.21 | 41.93 | 57.97 |
| | DMN [CVPR 2024] | ✗ | 46.28 | 45.44 | 76.82 | 75.32 | 42.71 | 57.31 |
| | TDA [CVPR 2024] | ✗ | 50.19 | 49.78 | 78.12 | 77.13 | 43.13 | 59.67 |
| | TPT [NeurIPS 2022] | ✓ | 49.71 | 48.89 | 74.82 | 73.39 | 43.10 | 57.98 |
| | DIffTPT [ICCV 2023] | ✓ | 49.45 | 49.19 | 74.98 | 74.31 | 42.93 | 58.17 |
| | RLCF [ICLR 2024] | ✓ | 40.53 | 39.79 | 71.21 | 69.63 | 31.77 | 50.59 |
| | **ML–TTA (Ours)** | ✓ | **52.92** | **52.24** | **78.72** | **78.13** | **43.62** | **61.13** |
| **ViT-B/32** | CLIP [ICML 2022] | ✓ | 50.31 | 50.15 | 77.18 | 76.85 | 42.90 | 59.48 |
| | DMN [CVPR 2024] | ✗ | 49.32 | 48.13 | 77.42 | 76.60 | 43.41 | 58.98 |
| | TDA [CVPR 2024] | ✗ | 51.23 | 51.49 | 77.62 | 77.12 | **44.13** | 60.32 |
| | TPT [NeurIPS 2022] | ✓ | 48.12 | 48.63 | 74.21 | 71.93 | 43.63 | 57.30 |
| | DIffTPT [ICCV 2023] | ✓ | 48.73 | 49.19 | 74.50 | 72.98 | 43.42 | 57.76 |
| | RLCF [ICLR 2024] | ✓ | 50.28 | 49.59 | 77.12 | 76.83 | 43.29 | 59.42 |
| | **ML–TTA (Ours)** | ✓ | **52.83** | **52.99** | **78.70** | **77.97** | 44.12 | **61.32** |
| **ViT-B/16** | CLIP [ICML 2022] | ✓ | 54.42 | 54.13 | 79.58 | 79.25 | 45.65 | 62.61 |
| | DMN [CVPR 2024] | ✗ | 52.52 | 52.37 | 79.83 | 79.67 | 46.27 | 62.13 |
| | DART [AAAI 2024] | ✗ | 54.73 | 54.68 | 79.91 | 78.56 | 45.91 | 62.76 |
| | TDA [CVPR 2024] | ✗ | 55.21 | 55.46 | 80.12 | 79.92 | **46.72** | 63.49 |
| | TPT [NeurIPS 2022] | ✓ | 53.32 | 54.20 | 77.54 | 77.39 | 46.15 | 61.72 |
| | DIffTPT [ICCV 2023] | ✓ | 53.91 | 54.15 | 77.93 | 77.24 | 46.13 | 61.87 |
| | RLCF [ICLR 2024] | ✓ | 54.21 | 54.43 | 79.29 | 79.26 | 43.18 | 62.07 |
| | **ML–TTA (Ours)** | ✓ | **57.52** | **57.49** | **81.28** | **81.13** | 46.55 | **64.80** |

For the vision transformer series architectures, compared to CLIP (Radford et al., 2021), ML–TTA consistently achieves $2 \sim 4\%$ mAP improvement on the COCO2014/2017 (Lin et al., 2014) and VOC2007/2012 (Everingham et al., 2010) datasets. However, most TTA methods, except TDA (Karmanov et al., 2024) and DART (Liu et al., 2024b), exhibit a slight performance decrement, particularly among episodic methods. Additionally, we observed an intriguing observation: all TTA methods, excluding RLCF (Zhao et al., 2024a), fail to substantially enhance the mAP performance of CLIP (Radford et al., 2021) on the NUSWIDE (Chua et al., 2009) dataset, with an improvement of merely about 1%. This may be attributed to the low image resolution of NUSWIDE dataset, where random data augmentation struggles to preserve sufficient visual information. Consequently, adapting to multi-labels for small targets may become a research topic in the future.

**Results on different prompt initialization.** For this comparison, we adopt the learned prompt from CoOp (Zhou et al., 2022b) and Maple (Khattak et al., 2023) to initialize the prompt weights, replacing the template "a photo of a **[CLS]**" employed in the original CLIP model. As shown in Table 2, the application of both CoOp and Maple prompt weights in our ML–TTA framework results in a significant enhancement of over 4% in mAP on the COCO2014/2017 datasets. For instance, the mAP increases from 47.53% to 51.58% and from 47.32% to 51.39% on COCO2014/2017 with CoOp prompt initialization, whereas other *episodic* methods, TPT (Shu et al., 2022) and DiffTPT (Feng et al., 2023), yield improvements of no more than 1.5%. Moreover, ML–TTA also surpasses

Table 2: Comparison with SOTAs on adapting multi-label instances with different prompt initialization.

| | Methods | *Epsdoic* | COCO2014 | COCO2017 | VOC2007 | VOC2012 | NUSWIDE | Average |
|---|---|---|---|---|---|---|---|---|
| **CoOp** | CoOp [IJCV2022] | ✓ | 56.12 | 56.35 | 79.14 | 77.85 | 46.74 | 63.24 |
| | TDA [CVPR 2024] | ✗ | 56.93 | 57.15 | 80.20 | 78.58 | 47.82 | 64.13 |
| | TPT [NeurIPS 2022] | ✓ | 55.35 | 55.23 | 79.72 | 77.85 | 47.27 | 63.08 |
| | DIffTPT [ICCV 2023] | ✓ | 55.30 | 55.47 | 79.86 | 77.61 | 47.13 | 63.07 |
| | RLCF [ICLR 2024] | ✓ | 56.72 | 56.18 | 80.15 | 78.24 | 47.62 | 63.78 |
| | **ML–TTA (Ours)** | ✓ | **59.68** | **59.33** | **83.17** | **81.36** | **48.12** | **66.33** |
| **Maple** | Maple [CVPR2023] | ✓ | 62.18 | 62.35 | 85.34 | 84.79 | 48.42 | 68.62 |
| | TDA [CVPR 2024] | ✗ | 63.25 | 63.64 | 85.76 | 84.15 | 49.55 | 69.27 |
| | TPT [NeurIPS 2022] | ✓ | 63.36 | 63.75 | 85.04 | 83.92 | 48.90 | 69.01 |
| | DIffTPT [ICCV 2023] | ✓ | 62.93 | 63.14 | 85.15 | 83.78 | 48.81 | 68.76 |
| | RLCF [ICLR 2024] | ✓ | 62.84 | 62.90 | 85.35 | 85.28 | 49.37 | 69.15 |
| | **ML–TTA (Ours)** | ✓ | **64.75** | **64.86** | **86.40** | **85.69** | **50.21** | **70.38** |

TDA (Karmanov et al., 2024), which is designed by dynamically employing the historical sample knowledge, on both CoOp and Maple prompt initialization across all datasets.

**Results on different label counts.** Apart from the analysis of architecture and prompt initialization weights, we explore a more challenging scenario in Table 3, where the COCO2014 dataset is divided into subsets with incrementally increasing numbers of image labels per part, *e.g.*, $\{1,2\}$ represents the number of labels $L$ per image is either 1 or 2. When $L \in \{1, 2\}$, TPT achieves only a neg-

Table 3: Results on different label counts.

| Methods | {1,2} | {3,4} | {5,6,7} | {>=8} |
|---|---|---|---|---|
| CLIP [ICML 2022] | 62.76 | 55.41 | 49.89 | 41.07 |
| TPT [NeurIPS 2022] | 62.88 | 53.05 | 45.57 | 37.43 |
| DiffTPT [ICCV 2023] | 61.97 | 52.67 | 44.32 | 36.89 |
| RLCF [ICLR 2024] | 66.01 | 51.65 | 43.32 | 35.08 |
| **ML–TTA (Ours)** | **67.14** | **57.59** | **51.68** | **41.32** |

ligible improvement compared to CLIP and shows large considerable performance degradation in other situations as well as DiffTPT. RLCF improves significantly when $L \in \{1, 2\}$, but its performance sharply declines as $L$ increases. In contrast, our ML–TTA framework outperforms CLIP across all situations, demonstrating that ML–TTA not only can address the distribution shifts during testing but also effectively handle varying numbers of labels in testing instances.

**Results on adaptation complexity.** Furthermore, we analyze adapting time per test instance with methods that also do not require retaining historical knowledge. Table 4 shows that ML–TTA presents a significant advantage compared to DiffTPT, which

Table 4: Results on adaptation complexity.

| Methods | TPT | DiffTPT | RLCF | ML-TTA |
|---|---|---|---|---|
| Adapting Time | **0.21**s | 0.41s | 0.45s | 0.24s |
| mAP | 48.52 | 48.56 | 36.87 | **51.58** |

involves generating multiple pseudo-images via a diffusion model, and RLCF, which requires distillation from a teacher model along with more gradient update steps. Compared to the benchmark TPT, ML-TTA increases adapting time due to simultaneous optimizing view and caption prompts.

## 4.3 ABLTION STUDIES.

**Different components.** In Table 5, we discuss the effectiveness of different components within our proposed ML–TTA framework on the COCO2014 and VOC2007 datasets, including view prompt (VP, *i.e.*, TPT (Shu et al., 2022)), caption prompt (CP), and Bound Entropy Minimization (BEM). Across both RN50 and ViT-B/16 architectures, BEM consistently

Table 5: Ablation studies of different components.

| Methods | RN50 | | ViT-B/16 | |
|---|---|---|---|---|
| | COCO2014 | VOC2007 | COCO2014 | VOC2007 |
| VP (*i.e.*, TPT) | 48.51 | 75.52 | 53.32 | 77.57 |
| VP+BEM | 48.96 | 76.31 | 53.58 | 77.89 |
| CP | 49.12 | 76.16 | 55.14 | 78.93 |
| CP+BEM | 49.54 | 76.75 | 55.64 | 79.58 |
| VP+CP | 51.22 | 77.98 | 57.14 | 80.85 |
| **VP+CP+BEM** | **51.58** | **78.62** | **57.52** | **81.28** |

enhances the mAP performance of VP, CP, and VP+CP, which indicates the reasonable effectiveness of our proposed Bound Entropy Minimization objective. Furthermore, we observe that CP and

CP+BEM always achieve superior performance compared to VP and VP+BEM in all settings. Such phenomenon shows treating text as a pseudo-image with a known label set to adapt multi-label test instance is more reliable than augmented views, as the positive label set of views is pseudo.

## 4.4 FURTHER ANALYSIS

Table 6: Comparison with binary cross-entropy loss.

| Methods | RN50 | | ViT-B/16 | |
|---|---|---|---|---|
| | COCO2014 | VOC2007 | COCO2014 | VOC2007 |
| CLIP | 47.53 | 75.91 | 54.42 | 79.58 |
| VP+CP+BCE | 48.39 | 75.75 | 54.51 | 78.59 |
| **VP+CP+BEM** | **51.58** | **78.62** | **57.52** | **81.28** |

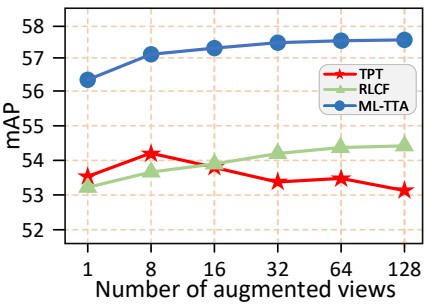

Figure 3: Results on different number of views.

**Loss functions.** Here, we conduct a comparison between Bound Entropy Minimization (BEM) and the conventional binary cross-entropy (BCE) loss function in multi-label classification tasks. Specifically, we regarded the *weak label set* of confident views as hard labels for those views and the *strong label set* of confident captions as hard labels for those captions, then using BCE loss to optimize the view and caption prompts. The results are shown in Table 6. Compared to CLIP, the mAP improvement using BCE loss on the COCO2014 is less than 1%. In contrast, our BEM objective surpasses BCE loss by 3~4% in mAP across all benchmarks, which demonstrates BEM is not only more effective than vanilla entropy minimization but also more robust compared to binary cross-entropy loss. BCE loss is not suitable for optimizing a single test instance.

**Number of augmented views.** Following TPT (Shu et al., 2022), we conduct parameter experiments of different numbers of augmented views on the COCO2014 dataset using ViT-B/16 architecture. As depicted in Figure 3, as the number of views increases from 1 to 128, the mAP performance of RLCF and ML–TTA both show an upward trend and begin to stabilize at 64 views. Surprisingly, the performance curve of TPT does not have any regularity, which implies that vanilla entropy minimization, by focusing only on the label with the highest probability, leads to unstable adaptation for multi-label instances.

Table 7: Results on different numbers of retrieved captions.

| Datasets | | CLIP | TPT | 1 | 2 | 4 | 8 | 16 | 32 | 64 | 128 |
|---|---|---|---|---|---|---|---|---|---|---|---|
| RN50 | COCO2014 | 47.53 | 48.52 | 51.35 | 51.37 | 51.41 | 51.49 | 51.58 | **51.59** | 51.55 | 51.48 |
| | VOC2007 | 75.91 | 75.54 | 78.29 | 78.33 | 78.48 | 78.54 | **78.61** | 78.59 | 78.53 | 78.42 |
| ViT-B/16 | COCO2014 | 54.42 | 53.32 | 57.23 | 57.33 | 57.41 | 57.48 | 57.49 | 57.52 | 57.55 | **57.58** |
| | VOC2007 | 79.58 | 77.54 | 81.06 | 81.12 | 81.21 | 81.24 | **81.28** | 81.19 | 81.15 | 80.98 |

**Number of retrieved captions.** We also investigate the impact of allocating different numbers of retrieved captions for each augmented view on the performance of ML–TTA. As shown in Table 7, when only one caption is allocated to each view, ML–TTA outperforms CLIP or TPT by 3~4%. As the number of captions increases, the performance of ML–TTA gradually improves until it stabilizes. For the VOC2007 dataset, too many captions can lead to a slight decrease in performance, as captions that are not highly similar to the view may introduce noisy positive labels that do not exist in the corresponding view.

## 5 CONCLUSION

In this paper, we investigate a test-time adaptation framework (ML–TTA) designed for multi-label data without making any presumptions about the distribution of the test instances. The proposed Bound Entropy Minimization (BEM) objective overcomes the limitation of the vanilla entropy loss, which only optimizes the most confident class. By conceptualizing paired captions as pseudo-views with a known label set, ML–TTA employs BEM to adapt to multi-label test instances by allocating *weak label set* to each augmented view and *strong label set* to each paired caption, binding the *top-k* predicted labels with the highest probabilities. Extensive experiments on the MSCOCO, VOC, and NUSWIDE datasets demonstrate that ML–TTA framework outperforms the source model CLIP and other state-of-the-art test-time adaptation methods, across various model architectures, prompt initialization, and varying label scenarios.

ACKNOWLEDGEMENTS

This work was supported in part by Key Laboratory of Target Cognition and Application Technology (2023-CXPT-LC-005).

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

# Appendix for Multi-Label Test-Time Adaptation with Bound Entropy Minimization

## A PROOF

### A.1 PROOF OF PROPOSITION 1

**Proposition 1.** Consider a model's output logits of a selected view $x$, denoted as $\mathbf{s} = (s_1, s_2, \ldots, s_L)$, where without loss of generality, we assume $s_1 > s_2 > \cdots > s_L$. It follows that the entropy loss $H = H(p(\cdot|x))$ decreases as $s_1$ increases, and $H$ increases as the sum of the remaining logits, $S_{\text{rest}} = \sum_{i=2}^{L} s_i$, decreases. Formally, this can be written as:

$$\frac{\partial H}{\partial s_1} < 0 \quad \text{and} \quad \frac{\partial H}{\partial S_{\text{rest}}} > 0. \tag{9}$$

*Proof.* We denote the predicted probability $p(y = l|x) = \frac{\exp s_l}{\sum_{i=1}^{L} \exp s_i}$ as $p_l$ for simplicity, where $s_i$ is the logit of the i-th category. We first calculate the partial derivative of $s_i$ with respect to $p_l$:

$$\begin{aligned}
\frac{\partial p_l}{\partial s_i} &= \frac{\partial}{\partial s_i} \left( \frac{\exp s_l}{\sum_{j=1}^{L} \exp s_j} \right) \\
&= \frac{\delta_{l,i} \exp s_l}{\sum_{j=1}^{L} \exp s_j} - \exp s_l \frac{\exp s_i}{\left( \sum_{j=1}^{L} \exp s_j \right)^2} \\
&= \delta_{l,i} p_l - p_l p_i
\end{aligned} \tag{10}$$

where $\delta_{i,j} = 1$ only if $i = j$, else is $\delta_{i,j} = 0$. We can now directly calculate the partial derivative of $H$ for $s_i$.

$$\begin{aligned}
\frac{\partial H}{\partial s_i} &= \frac{\partial}{\partial s_i} \left( -\sum_{l=1}^{L} p_l \log p_l \right) \\
&= -\sum_{l=1}^{L} \left( \frac{\partial p_l}{\partial s_i} \log p_l + p_l \frac{1}{p_l} \frac{\partial p_l}{\partial s_i} \right) \\
&= -\sum_{l=1}^{L} \left( \delta_{l,i} p_l \log p_l - p_l p_i \log p_l + \delta_{l,i} p_l - p_l p_i \right) \\
&= p_i \log p_i + p_i - \sum_{l=1}^{L} \left( -p_l p_i \log p_l - p_l p_i \right) \\
&= (p_i \log p_i + p_i) \left( \sum_{l=1}^{L} p_l \right) - \sum_{l=1}^{L} \left( -p_l p_i \log p_l - p_l p_i \right) \\
&= -\sum_{l=1}^{L} \left( p_l p_i \log p_i - p_l p_i \log p_l + p_l p_i - p_l p_i \right) \\
&= -\sum_{l=1}^{L} p_l p_i \log \frac{p_i}{p_l}
\end{aligned} \tag{11}$$

where the fourth equivalent uses the property of $\delta_{i,j}$ and fifth equivalent uses $\sum_{l=1}^{L} p_l = 1$. Since we assume $s_1 > s_2 > \cdots > s_L$, then the probabilities have the same order $p_1 > p_2 > \cdots > p_L$, therefor:

$$\frac{\partial H}{\partial s_1} = -\sum_{l=1}^{L} p_l p_1 \log \frac{p_1}{p_l} = -\sum_{l=2}^{L} p_l p_1 \log \frac{p_1}{p_l} < 0 \tag{12}$$

as $\log \frac{p_1}{p_l} > 0$ for all $l > 1$, therefor we proof the first inequality in proposition 1. To prove the second inequality, we first calculate the sum of the partial derivative of $H$ for all logits.

$$
\begin{aligned}
\sum_{i=1}^{L} \frac{\partial H}{\partial s_i} &= \sum_{i=1}^{L} \left( -\sum_{l=1}^{L} p_l p_i \log \frac{p_i}{p_l} \right) \\
&= -\sum_{i=1}^{L} \sum_{l=1}^{L} (p_l p_i \log p_i - p_l p_i \log p_l) \\
&= -\sum_{i=1}^{L} \sum_{l=1}^{L} (p_l p_i \log p_i - p_i p_l \log p_i) \\
&= 0
\end{aligned}
\tag{13}
$$

where we change the position of index $i$ and $l$ for the second term in the double summation to get the third equivalent. Now the second inequality is easy to get:

$$
\begin{aligned}
\frac{\partial H}{\partial S_{\text{rest}}} &= \sum_{i=2}^{L} \frac{\partial H}{\partial s_i} \bigg/ \frac{\partial S_{\text{rest}}}{\partial s_i} \\
&= \sum_{i=2}^{L} \frac{\partial H}{\partial s_i} \bigg/ 1 \\
&= \sum_{i=1}^{L} \frac{\partial H}{\partial s_i} - \frac{\partial H}{\partial s_1} \\
&= -\frac{\partial H}{\partial s_1} > 0
\end{aligned}
\tag{14}
$$

## A.2 PROOF OF PROPOSITION 2

*proposition 2. Consider a model's output logits of a selected view $x$, denoted as $\mathbf{s} = (s_1, s_2, \ldots, s_L)$, where without loss of generality, we assume $s_1 > s_2 > \cdots > s_L$. We define the modified logits as $\mathbf{s}' = (s'_1, s'_2, \ldots, s'_L)$, where $s'_i = a_i + s_i$ for $i \leq k$ with $a_i = s_1 - s_i$ and $s'_i = s_i$ for $i > k$. Here, $a_i$ is a constant that does not participate in differentiation, resulting in $s'_i = s_1$ for all $i \leq k$. Let $S_{rest} = \sum_{i=k+1}^{L} s_i$. For the modified logits $\mathbf{s}'$, we define the modified probability $\mathbf{p}' = Softmax(\mathbf{s}')$, and the modified entropy as $H' = -\sum_{i=1}^{L} p'_i \log p'_i$. It follows that:*

$$
\frac{\partial H'}{\partial s_i} < 0, \quad \forall i \leq k \quad \text{and} \quad \frac{\partial H'}{\partial S_{\text{rest}}} > 0.
\tag{15}
$$

*Proof.* With the assumption $s_1 > s_2 > \cdots > s_L$ and the definition of $\mathbf{s}'$, we have $s'_1 = s'_2 = \cdots = s'_k > \cdots > s'_L$. Use the conclusion in proposition 1, for $i \leq k$, we have:

$$
\frac{\partial H'}{\partial s_i} = \frac{\partial H'}{\partial s'_i} \frac{\mathrm{d}s'_i}{\mathrm{d}s_i} = \frac{\partial H'}{\partial s'_i} \times 1 = -\sum_{l=1}^{L} p'_l p'_i \log \frac{p'_i}{p'_l} = -\sum_{l=k+1}^{L} p'_l p'_i \log \frac{p'_i}{p'_l} < 0 \\
, \quad \forall i \leq k
\tag{16}
$$

where $p'_i > p'_l$ for $i \leq k$ and $l > k$ as $s'_1 = s'_2 = \cdots = s'_k > \cdots > s'_L$. Similar to the proof of proposition 1, we use the conclusion of $\sum_{i=1}^{L} \frac{\partial H}{\partial s'_i} = 0$, which has been proved in Equation. 13 to

prove the second inequality.

$$
\begin{aligned}
\frac{\partial H'}{\partial S_{\text{rest}}} &= \sum_{i=k+1}^{L} \frac{\partial H}{\partial s_i} \Big/ \frac{\partial S_{\text{rest}}}{\partial s_i} \\
&= \sum_{i=k+1}^{L} \frac{\partial H}{\partial s_i} \Big/ 1 \\
&= \sum_{i=1}^{L} \frac{\partial H}{\partial s_i} - \sum_{i=1}^{k} \frac{\partial H}{\partial s_i} \\
&= -\sum_{i=1}^{k} \frac{\partial H}{\partial s_i} > 0
\end{aligned}
\tag{17}
$$

## B    DETAILED LABEL BINDING PROCESS.

In this section, we present a certain example to showcase the calculation of Label Binding 3.3.2. Label binding refers to making the *top-k* predicted logits equal, as expressed below:

$$
\tilde{s}_{ij}^{\mathbf{x}^{\text{test}}} = ((m_i^{\mathbf{x}^{\text{test}}} - s_{ij}^{\mathbf{x}^{\text{test}}}) + s_{ij}^{\mathbf{x}^{\text{test}}}) \times \mathbb{I}(\text{Rank}_-(s_{ij}^{\mathbf{x}^{\text{test}}}, \mathbf{s}_i^{\mathbf{x}^{\text{test}}}) \le k^{\mathbf{x}^{\text{test}}}) + s_{ij}^{\mathbf{x}^{\text{test}}} \times \mathbb{I}(\text{Rank}_-(s_{ij}^{\mathbf{x}^{\text{test}}}, \mathbf{s}_i^{\mathbf{x}^{\text{test}}}) > k^{\mathbf{x}^{\text{test}}}), \tag{18}
$$

Since label binding (making ... equal) is non-differentiable, we employ the stop-gradient operation in VQ-VAE van den Oord et al. (2017) for backpropagation, *i.e.* $((m_i^{\mathbf{x}^{\text{test}}} - s_{ij}^{\mathbf{x}^{\text{test}}}) + s_{ij}^{\mathbf{x}^{\text{test}}})$ to perform label binding.

Taking a 3-class classification task as an example with class labels of $(1, 2, 3)$, assuming $k^{\mathbf{x}_i^{\text{test}}}$ is 2, and the label binding process is $\mathbf{s} = [\mathbf{0.9}, \mathbf{0.7}, 0.3] \rightarrow \mathbf{s}' = [\mathbf{0.9}, \mathbf{0.9}, 0.3]$. $\tilde{s}_{ij}^{\mathbf{x}^{test}}$ represents the logit of the $j$-th class in the $i$-th augmented view after label binding, *e.g.*, $\tilde{s}_{i2}^{\mathbf{x}^{test}}$ changes from $\mathbf{0.7} \rightarrow \mathbf{0.9}$. $m_i^{\mathbf{x}^{\text{test}}}$ denotes the maximum value of $\mathbf{s}$, which is $\mathbf{0.9}$. $\mathbb{I}(\cdot)$ is the indicator function. $\text{Rank}_-(a, \mathbf{b})$ indicates the descending rank of $a$ within $\mathbf{b}$, *e.g.*, $\text{Rank}_-(0.7, \mathbf{s}) = 2$. The process for computing the bound logit for each class is as follows:

$$
\begin{aligned}
\tilde{s}_{i1}^{\mathbf{x}^{\text{test}}} &= ((0.9 - 0.9) + 0.9) \times \mathbb{I}(\text{Rank}_-(0.9, \mathbf{s}) \le 2) + 0.9 \times \mathbb{I}(\text{Rank}_-(0.9, \mathbf{s}) > 2) \\
&= 0.9 \times \mathbb{I}(1 \le 2) + 0.9 \times \mathbb{I}(1 > 2) \\
&= 0.9, \\
\tilde{s}_{i2}^{\mathbf{x}^{\text{test}}} &= ((0.9 - 0.7) + 0.7) \times \mathbb{I}(\text{Rank}_-(0.7, \mathbf{s}) \le 2) + 0.7 \times \mathbb{I}(\text{Rank}_-(0.7, \mathbf{s}) > 2) \\
&= 0.9 \times \mathbb{I}(2 \le 2) + 0.7 \times \mathbb{I}(2 > 2) \\
&= 0.9, \\
\tilde{s}_{i3}^{\mathbf{x}^{\text{test}}} &= ((0.9 - 0.3) + 0.3) \times \mathbb{I}(\text{Rank}_-(0.3, \mathbf{s}) \le 2) + 0.3 \times \mathbb{I}(\text{Rank}_-(0.3, \mathbf{s}) > 2) \\
&= 0.9 \times \mathbb{I}(3 \le 2) + 0.3 \times \mathbb{I}(3 > 2) \\
&= 0.3,
\end{aligned}
\tag{19}
$$

label binding process changes the logits from $[\mathbf{0.9}, \mathbf{0.7}, 0.3] \rightarrow [\mathbf{0.9}, \mathbf{0.9}, 0.3]$.

## C    TEXT DESCRIPTION BASE CONSTRUCTION

Here, we present the construction of the text description base using Large language models (LLMs). Initially, for a set of labels, denoted as $\mathcal{L} = \{l_1, l_2, ..., l_L\}$, where $L$ represents the total number of labels across all multi-label datasets. Following PVP (Wu et al., 2024), we define a prompt template to instruct LLama-2-7B (Touvron et al., 2023), generating descriptions that describe a nature scene, which is as follows:

*PROMPT: Make a sentence to describe a photo. Requirements: Each sentence should be less than 15 words and include keywords: $\{l_{i_1}, l_{i_2}, \ldots, l_{i_j}\}$,*

where $\{l_{i_1}, l_{i_2}, \ldots, l_{i_j}\}$ is a subset of $\mathcal{L}$ with $i \le 5$. We randomly sample $j$ categories from $\mathcal{L}$ and input these categories along with the prompt template into LLMs to automatically generate text

descriptions. After obtaining generated descriptions, we employ the nouns filtering strategy used in PVP to extract textual labels for each description. Some examples are illustrated below:

1. A hot dog toaster is positioned next to a stop sign.

2. A group of girls enjoying a game of frisbee while sitting on chairs.

3. The little boy dreams of becoming a pilot as he falls asleep with his aeroplane.

4. Remotes control the TV, allowing people to enjoy their favorite shows.

5. A motorbike speeds past a man wearing a tie, as he holds a wine glass in one hand.

where the underlined words indicate the textual labels extracted from the corresponding description. However, due to the uncontrollable quality and relevance of the paired captions generated by LLMs, these captions may not always accurately represent the image contents. In real-world scenarios, besides adopting a confident-based filtering strategy to filter out views and captions with high entropy (*e.g.*, low confidence), we can also explore more robust strategies to retrieve paired captions, such as, constructing high-quality and content-rich text description databases, ensembling label sets from multiple captions, or improving the similarity retrieval strategy, thereby reducing the impact of noise on the model's adaptation.

