# OpenReview forum: "Multi-Label Test-Time Adaptation with Bound Entropy Minimization"
_ICLR.cc/2025/Conference — ICLR 2025 Poster_

### Official Review · Reviewer_LTFP · 2024-11-01

**Soundness:** 3
**Presentation:** 3
**Contribution:** 3
**Rating:** 8
**Confidence:** 3

**Summary:**

The paper presents a novel approach to Test-Time Adaptation (TTA) for multi-label scenarios using a method termed Bound Entropy Minimization (BEM). The paper is well-structured, the problem statement is clear, and the proposed solution is innovative. The integration of view and caption prompts and the application of BEM to meet the test time adaptation are innovative to some extent. However, there are  some details should be clarified.

**Strengths:**

1) This paper is well-structured, the problem statement is clear, and the proposed solution is innovative.
2) The integration of view and caption prompts and the application of BEM to meet the test time adaptation are innovative to some extent.
3) Compared with the latest and most advanced methods, the method in this paper achieves the best performance.

**Weaknesses:**

1) In your paper, the choice of top-k seems to be very important, so how do you determine the setting of k? You said "we retrieve a paired caption with derived textual labels for each view, which then serves as weak label set of size k for the corresponding view." How do you make sure the selected weak label set is reliable?
2) I can not see any explanation about the "augmented view" in this paper, what is the definition of it and what effort does it have in the framework?
3) The comparison methods you selected in the paper may be not designed for multi-label datasets, so is this comparison fair? Could you add more ML-TTA specific framework to the results?
4) Some details: Table 1 lacks a description of evaluation metric; Marking the second-best result in the experimental results is more beneficial to the reader.

**Questions:**

See the weaknesses

---

> ### Author Response · Authors · 2024-11-18
> **Official Response by Authors (1/1)**
>
> We thanks for your valuable suggestions and we will try to address your concerns as follows. We are eager to engage in a more detailed discussion with you.
> ### **Weakness 1: Determine the setting of k. How to make sure the weak label set is reliable?**
> 1.Determine the setting of k.
> + K is determined based on the number of textual labels contained in the paired caption, it is not a hyperparameter. For each augmented view $\mathbf x_i$, we retrieve the most similar caption $\mathbf t_i$ for $\mathbf x_i$. For a specific pair $\langle \mathbf x_i, \mathbf t_i\rangle$, assuming $\mathbf t_i$ is **A black Honda bicycle parked in front of a car**, we follow the noun filtering in PVP [1] to extract the label set **{bicycle, car}** from $\mathbf t_i$ with the size of 2.
> + This label set serves as both the strong label set for the caption and the weak label set for the view, hence the value of **k** is 2. If $\mathbf t_i$ is **A group of girls enjoying a game of frisbee while sitting on chairs**, the label set would be **{girls, frisbee, chairs}**, and the value of **k** would be 3.
>
> 2.How to make sure ... is reliable.
> + Captions primarily describe salient visual information in the image and may not accurately reflect smaller object categories within the image. Therefore, for the caption **“A black Honda bicycle parked in front of a car”**, we refer to the label set **{bicycle, car}** as the **strong** label set for the caption as it is directly extracted from the caption. Likewise, it also serves as the **pseudo labels** for the view, $i.e.$, **weak** label set.
> + Moreover, we aim to build a TTA framework for multi-label scenarios, demonstrating the feasibility of traditional entropy minimization methods in multi-label instances. In practical applications, we can consider employing a more robust similarity retrieval strategy, integrating label sets from multiple captions, or constructing a more comprehensive text description base to enhance the reliability of the weak label set.
>
> ### **Weakness 2: Explanation of "augmented view".**
> + We apologize for the confusion of the definition. Augmented view is a widely adopted method in the TTA domain, which involves generating a set of $N$ different views through data augmentations. Then, TTA selects the **top 10%** highest confidence views and minimizes the marginal entropy of these views, encouraging consistent and confident predictions.
> + Our work follows the same method in the multi-label TTA scenario, performing $N$ data augmentations on multi-label instances and optimizing the view and caption prompts with the proposed Bound Entropy Minimization to enhance the consistency of model predictions.
> ### **Weakness 3: Add ML-TTA specific framework for comparison.**
> + Currently, TTA methods primarily focus on multi-class scenarios, adapting single-label instances through entropy minimization. However, for multi-label instances, considering only the **top-1** class inevitably harms the prediction performance of other positive labels.
> + To our knowledge, our work is the first to explore the feasibility of entropy minimization in multi-label scenarios. The proposed Bound Entropy Minimization (BEM) aims to simultaneously increase the confidence of multiple **top-k** labels. Therefore, we select the SOTA methods in the TTA field for multi-class scenarios as benchmarks, such as RCLF [2] and TDA [3]. Moreover, our work demonstrates the feasibility of entropy minimization in multi-label TTA and provides a basic framework for subsequent multi-label TTA tasks.
> ### **Weakness 4: Explanation of evaluation metric in Table 1. Marking the second-best result in the experiments.**
> + The evaluation metric in Table 1 is the widely used **mean average precision (mAP)** in multi-label classification tasks. mAP is the average of **Average Precision (AP)**, where AP is the area under the **Precision-Recall curve**. Precision is the proportion of truly positive samples among all samples predicted as positive by the model, and Recall is the proportion of truly positive samples that are correctly predicted as positive by the model.
>
> + In multi-label classification, for each category, we can draw a Precision-Recall curve and calculate the area under this curve, which is the AP for that category. Therefore, the calculation steps for mAP are: Calculate the AP for each category. Take the average of all category AP values to get mAP. The mAP is computed as follows: $mAP=\frac{1}{L}\sum_{i=1}^L AP_i$, where $L$ is the number of categories, and $AP_i$ is the average precision for the **i-th** category.
>
> + We will introduce mAP in detail and mark the second-best result in the experiments in the future version.
>
> [1]. TAI++:Text as Image for Multi-Label Image Classification by Co-Learning Transferable Prompt. IJCAI 2024
>
> [2]. Test-Time Adaptation with CLIP Reward for Zero-Shot Generalization in Vision-Language Models. ICLR 2024
>
> [3]. Efficient Test-Time Adaptation of Vision-Language Models. CVPR 2024

---

> > ### Comment · Reviewer_LTFP · 2024-11-24
> > **Thanks for your feedback**
> >
> > Some of my concerns are addressed by authors. As for the experiments that you can not add, I still have reservations about it.

---

> ### Author Response · Authors · 2024-11-23
> **Looking forward to your feedback.**
>
> Dear Reviewer, we are looking forward to your professional suggestions and hope to receive your guidance to further discuss and refine the contents of the work. We are eager for your response. Thank you.

---

> ### Author Response · Authors · 2024-11-25
> **Reply to Reviewer LTFP**
>
> + We appreciate your careful review of our work. Throughout the research of ML-TTA, we conducted extensive investigation and discourse, including a large number of papers and surveys [1][2][3][4] on TTA. To our knowledge, our work is the first study to examine TTA within multi-label scenarios.
> + To facilitate a relatively fair comparison, in our experiments, we selected and adjusted current SOTA methods for multi-class TTA to the multi-label scenarios. Following your suggestion, we further added and adjusted more latest multi-class TTA methods for comparison on ViT-B/16 architecture as below, highlighting the advantages of ML-TTA equipped with Bound Entropy Minimization (BEM) in multi-label scenarios. It is indicated that ML-TTA achieves the best performance across all benchmarks. Even though these methods employ innovative strategies such as class prototypes, optimal transport, and bias correction, their performance still does not significantly outperform CLIP in the multi-label scenarios.
>
> | Methods      | COCO2014 | COCO2017 | VOC2007 | VOC2007 | NUSWIDE | Average |
> | ------- | ------- | ------- | ------- | ------- | ------- | ------- |
> | CLIP         | 54.42 | 54.13 | 79.58 | 79.25 | 45.65 | 62.61 |
> | DPE-CLIP [5] | 54.86 | 54.71 | 80.05 | 79.55 | 45.32 | 62.89 |
> | AWT [6]      | 54.95 | 55.10 | 79.86 | 79.47 | 45.63 | 63.00 |
> | ZERO [7]   | 55.12 | 54.92 | 79.94 | 79.75 | 45.58 | 63.06 |
> | **ML-TTA**   | **57.52** | **57.49** | **81.28** | **81.13** | **46.55** | **64.80** |
>
> + Our proposed Bound Entropy Minimization (BEM) explores the feasibility of the TTA paradigm in multi-label scenarios, and we hope that the innovation of ML-TTA will attract the attention of more researchers and inspire more excellent works in multi-label TTA. Once again, thank you for your valuable feedback, and we eagerly await your further guidance.
>
> [1].A comprehensive survey on test-time adaptation under distribution shifts. IJCV 2024
>
> [2].A comprehensive survey on source-free domain adaptation. TPAMI 2024
>
> [3].In search of lost online test-time adaptation: A survey. IJCV 2024
>
> [4].Beyond model adaptation at test time: A survey. arXiv 2024
>
> [5].Dual Prototype Evolving for Test-Time Generalization of Vision-Language Models. NeurIPS 2024
>
> [6].AWT: Transferring Vision-Language Models via Augmentation, Weighting, and Transportation. NeurIPS 2024
>
> [7].Frustratingly Easy Test-Time Adaptation of Vision-Language Models. NeurIPS 2024

---

> ### Author Response · Authors · 2024-11-30
> **Looking forward to your feedback**
>
> Dear reviewer LTFP, thanks for your previous suggestions for our work. We would like to further discuss the content with you and hope to receive your response to the manuscript.
>
> Additionally, if you find that the overall quality of the manuscript has improved after re-evaluating these modifications, we kindly ask you to consider adjusting the rating score accordingly.
>
> Looking forward to your feedback, thank you!

---

> ### Author Response · Authors · 2024-12-03
>
> Dear reviewer LTFP,
>
> Thanks again for your previous feedback. We wish to discuss the manuscript content with you and hope for your response.
>
> If you find the manuscript’s quality improved, we kindly request you to consider revising the rating score.
>
> Best regards,

---

> > ### Comment · Reviewer_LTFP · 2024-12-03
> > **Thanks for response!**
> >
> > I hope the updated content could appear in the final version upon the acceptance of this paper and the code will be open-sourced. I will raise the score, thanks!

---

> > > ### Author Response · Authors · 2024-12-03
> > > **Thanks for response and time!**
> > >
> > > Thanks for your response and time, as well as your suggestions for this work~

---

### Official Review · Reviewer_bCKi · 2024-11-01

**Soundness:** 4
**Presentation:** 3
**Contribution:** 3
**Rating:** 6
**Confidence:** 4

**Summary:**

This paper introduces a Bound Entropy Minimization method for improving test-time adaptation in multi-label scenarios. BEM addresses the challenge of adapting multiple labels simultaneously. By integrating textual captions to determine the number of positive labels, the method enhances the confidence of several top predicted labels. The proposed Multi-Label Test-Time Adaptation (ML–TTA) framework leverages both visual and textual data, leading to superior performance across various datasets compared to state-of-the-art techniques.

**Strengths:**

1. The proposed Bound Entropy Minimization (BEM) method presents an innovative solution to improve test-time adaptation in multi-label scenarios.
2. The use of paired captions as pseudo-labels is a clever strategy to determine the number of positive labels for each test instance.
3.  It considers both visual and textual modalities, optimizing for a more robust adaptation to distribution shifts.
4. The figures are well presented.

**Weaknesses:**

1. More detailed motivation behind the model design is preferred. It is important to explain why the authors propose the method in this work.
2. The proposed method involves multiple steps, including view augmentation, caption retrieval, and label binding, which might introduce complexity in practical implementation. Simplifying the process could enhance usability.
3. The effectiveness of the method heavily relies on the quality and relevance of the paired captions. In real-world scenarios, captions might not always accurately represent the image content, which could affect performance.

**Questions:**

Please refer to the weakness.

---

> ### Author Response · Authors · 2024-11-18
> **Official Response by Authors (1/1)**
>
> Thanks for your valuable suggestions, we will try to address your concerns and we are eager to engage in a more detailed discussion with you.
> ### **Weakness 1: More detailed motivation about ML-TTA.**
> + Test-time adaptation (TTA) refers to directly adapting test instances without the need to access original training data. However, existing TTA methods are based on entropy minimization and primarily focus on increasing the predicted confidence of the **top-1** label. In multi-label scenarios, however, optimizing only the **top-1** label may result in insufficient adaptation for other positive labels.
>
> + To address this issue, we propose the Boundary Entropy Minimization (BEM) objective, which aims to simultaneously increase the confidence of multiple **top-k** labels, where **k** is determined by the retrieved paired caption. The core idea of BEM involves treating the weak label set of each augmented view and the corresponding strong label set of each caption as single-label, learning instance-level view and caption prompts to adapt to multi-label instances. By binding the **top-k** predicted labels, BEM mitigates the limitation of traditional entropy minimization and avoids over-optimizing the **top-1** label.
>
> ### **Weakness 2: ML-TTA may introduce complexity in practice.**
> + ML-TTA consists of: view augmentation, caption retrieval, and label binding.
>
>     + View augmentation is a widely adopted method in the TTA domain, which encourages the model to make consistent and confident predictions by minimizing the marginal entropy of predictions across multiple views.
>
>     + For caption retrieval, ML-TTA pre-constructs an offline embedding base of text descriptions. Hence, in practice, only a single matrix multiplication is sufficient to retrieve the paired captions.
>
>     + Label binding refers to making the logits of the **top-k** labels equal, as expressed by Eq.(6) in the manuscript: $ \tilde s_{ij}^{\mathbf x^{\text {test}}}=( ( m_i^{\mathbf x^{\text {test}}} - s_{ij}^{\mathbf x^{\text {test}}} ) + s_{ij}^{\mathbf x^{\text {test}}} ) \times \mathbb I( \mathrm {Rank}\_{( s_{ij}^{\mathbf x^{\text {test}}}, \mathbf s_{i}^{\mathbf x^{\text {test}}})} \leq k^{\mathbf x_i^{\text {test}}} )+ s_{ij}^{\mathbf x^{\text {test}}} \times \mathbb I (\mathrm {Rank}\_{(s_{ij}^{\mathbf{x}^{\text {test}}}, \mathbf s_i^{\mathbf x^{\text {test}}})} > k^{\mathbf x_i^{\text {test}}} )$. We can see that label binding involves some simple mathematical operations and stop-gradient operations $( ( m_i^{\mathbf x^{\text {test}}} - s_{ij}^{\mathbf x^{\text {test}}} ) + s_{ij}^{\mathbf x^{\text {test}}} )$, which only need a negligible time consumption during the model adaptation process.
> + Furthermore, we conduct analysis of testing time per test instance on the MSCOCO-2014 dataset, comparing ML-TTA with others that also do not require retaining historical knowledge, as shown in the Table below:
> | Methods           | TPT [1] | DiffTPT [2] | RLCF [3] | **ML-TTA**|
> | ------- | ------- | ------- | ------- | ------- |
> | **Testing Time**      | $\mathbf {0.21}s$ | $0.41s$ | $0.45s$ | $\underline {0.24}s$ |
> | **mAP**               | $48.52$ | $\underline {48.56}$ | $36.87$ | $\mathbf {51.58}$ |
> + The result shows that compared to the benchmark TPT, ML-TTA exhibits an increase in testing time due to the simultaneous optimization of view and caption prompts. However, ML-TTA presents a significant advantage compared to DiffTPT, which involves generating multiple pseudo-images via a diffusion model, and RLCF, which requires distillation from a teacher model along with more gradient update steps.
>
> ### **Weakness 3: In real-world scenarios, captions may not always accurately represent the image content.**
> + Our work employs captions to determine the number of labels for views. Even if there is some deviation between captions and contents of views, the proposed BEM objective can effectively mitigate the limitation of traditional entropy minimization that only optimizes for the **top-1** label.
>
> + Indeed, in real-world application scenarios, the accuracy of retrieved paired captions may be affected by various factors. To address this, in the manuscript, we also adopt a confident-based filtering strategy, filtering out views and captions with high entropy ($i.e.$, low confidence) to reduce the impact of noise on the model's adaptation.
>
> + Furthermore, we can explore more robust strategies to retrieve paired captions in future works, such as, constructing high-quality and content-rich text description databases, ensembling label sets from multiple captions, or improving the similarity retrieval strategy.
>
> [1]. Test-Time Prompt Tuning for Zero-Shot Generalization in Vision-Language Models. NeurIPS 2022
>
> [2]. Diverse Data Augmentation with Diffusions for Effective Test-time Prompt Tuning. ICCV 2023
>
> [3]. Test-Time Adaptation with CLIP Reward for Zero-Shot Generalization in Vision-Language Models. ICLR 2024

---

> > ### Comment · Reviewer_bCKi · 2024-11-25
> >
> > Thanks for the authors' responses. The authors have addressed my concerns and I will maintain my score.

---

> > > ### Author Response · Authors · 2024-11-25
> > > **Response**
> > >
> > > Thanks for your time and response~

---

> ### Author Response · Authors · 2024-11-23
> **Looking forward to your feedback.**
>
> Dear Reviewer, we are looking forward to your professional suggestions and hope to receive your guidance to further discuss and refine the contents of the work. We are eager for your response. Thank you.

---

> ### Author Response · Authors · 2024-11-25
> **Looking forward to your feedback**
>
> Dear Reviewer, your professional suggestions are crucial to our research. We kindly ask that you respond at your earliest convenience to further discuss and refine our work. We are eagerly awaiting your valuable feedback. Thank you!

---

### Official Review · Reviewer_97yB · 2024-11-02

**Soundness:** 2
**Presentation:** 3
**Contribution:** 2
**Rating:** 6
**Confidence:** 5

**Summary:**

This paper proposes a novel method for Multi-Label Test-Time Adaptation (ML–TTA) using a technique called Bound Entropy Minimization (BEM). Unlike traditional test-time adaptation (TTA) that optimizes for the most confident single-label prediction, BEM increases the confidence of the top-k predicted labels simultaneously. This approach addresses the challenges associated with multi-label data where prioritizing one label can reduce the adaptation effectiveness for others. The framework also incorporates paired captions as pseudo-positive labels to guide adaptation. Experiments conducted on MSCOCO, VOC, and NUSWIDE datasets demonstrate that ML–TTA outperforms existing methods and the original CLIP model, showcasing superior adaptability across diverse architectures and prompt setups.

**Strengths:**

1. The paper demonstrates robust experimentation across diverse datasets (MSCOCO, VOC, NUSWIDE) and architectures (e.g., RN50, ViT-B/16), showcasing the generalizability and efficacy of the proposed method.
2. The introduction of the Bound Entropy Minimization (BEM) for Multi-Label Test-Time Adaptation (ML–TTA) is a significant theoretical and practical advancement. It effectively addresses the challenges inherent in multi-label test-time adaptation, a space where traditional single-label approaches like entropy minimization fall short.

**Weaknesses:**

1. The method section, particularly the mathematical formulations and algorithmic details, could be more clearly presented. The explanations surrounding the implementation of label binding and how the paired captions are retrieved need additional clarity for readers less familiar with the intricate mechanisms of vision-language model adaptations.
2. While the paper effectively shows ML–TTA's superiority over traditional methods, it would benefit from a more detailed discussion about the choice of baseline methods and potential reasons for their relative underperformance.

**Questions:**

See weakness.

---

> ### Author Response · Authors · 2024-11-18
> **Official Response by Authors (1/2)**
>
> Thanks for your valuable suggestions, we will try to address your concerns and we are eager to engage in a more detailed discussion with you.
> ### **Weakness 1: Explanation of paired captions retrieval, label binding, and algorithmic details.**
> 1.Paired captions retrieval.
> + Given a test image $\mathbf x$, $\mathbf x$ is first augmented $N$ times to obtain a set of different views {$\mathbf x_i|i=1,2,3,...,N$}. The goal of paired caption retrieval is to retrieve the most similar caption for each view. Initially, we collect massive text descriptions following PVP [1]. Then, CLIP is used to extract text embeddings and construct an offline database of size $B\times d$, where $B$ denotes the number of test descriptions and $d$ denotes the embedding dimension.
> + For a given augmented view $\mathbf x_i$, defined as a $d$-dimensional vector, we directly compute the similarity between $\mathbf x_i$ and all text embeddings in the database, resulting in a $B$-dimensional similarity vector. The text description corresponding to the highest similarity is considered as the retrieved paired caption for $\mathbf x_i$.
>
> 2.Label binding.
> + Bound Entropy Minimization (BEM) aims to simultaneously increase the prediction confidence for the **top-k** labels, whereas the traditional entropy minimization can only enhance the confidence of the **top-1** label. Proposition 2 in the manuscript states that the key point in BEM is to equalize the logits of the **top-k** labels, $i.e.$, label binding process. The Eq.(6) in the manuscript is as follows:
>
>     $ \tilde s_{ij}^{\mathbf x^{\text {test}}}=( ( m_i^{\mathbf x^{\text {test}}} - s_{ij}^{\mathbf x^{\text {test}}} ) + s_{ij}^{\mathbf x^{\text {test}}} ) \times \mathbb I( \mathrm {Rank}\_{( s_{ij}^{\mathbf x^{\text {test}}}, \mathbf s_{i}^{\mathbf x^{\text {test}}})} \leq k^{\mathbf x_i^{\text {test}}} )+ s_{ij}^{\mathbf x^{\text {test}}} \times \mathbb I (\mathrm {Rank}\_{(s_{ij}^{\mathbf{x}^{\text {test}}}, \mathbf s_i^{\mathbf x^{\text {test}}})} > k^{\mathbf x_i^{\text {test}}} )$.
>
> + We take a **3-class** classification task with class labels of **(1,2,3)** as an example, assuming $k^{\mathbf x_i^{\text {test}}}$ is **2**, and the label binding process is $\mathbf s = [\mathbf {0.9}, \mathbf {0.7}, 0.3] \rightarrow \mathbf s^{'} = [\mathbf {0.9}, \mathbf {0.9}, 0.3]$. $\tilde s_{ij}^{\mathbf x^{\text test}}$ represents the **logit** of the **j-th** class in the **i-th** augmented view after label binding, $e.g.$, $\tilde s_{i2}^{\mathbf x^{\text test}}$ changes from $\mathbf {0.7}\rightarrow\mathbf{0.9}$. $m_i^{\mathbf x^{\text {test}}}$ denotes the maximum value of $\mathbf s$, which is $\mathbf {0.9}$. $\mathbb I(\cdot)$ is the indicator function. $\mathrm {Rank}\_{(a, \mathbf b)}$ indicates the descending rank of $a$ within **b**, $e.g.$, $\mathrm {Rank}\_{(0.7, \mathbf s)} = 2$. The process for computing the **bound logit** for each class is as follows:
>
>     $ \tilde s_{i1}^{\mathbf x^{\text {test}}}=( ( 0.9 - 0.9 ) + 0.9 ) \times \mathbb I( \mathrm {Rank}\_{( 0.9, \mathbf s)} \leq 2)+ 0.9 \times \mathbb I (\mathrm {Rank}\_{(0.9, \mathbf s)} > 2) = 0.9 \times \mathbb I( 1 \leq 2)+ 0.9 \times \mathbb I (1 > 2) = 0.9$
>
>     $ \tilde s_{i2}^{\mathbf x^{\text {test}}}=( ( 0.9 - 0.7 ) + 0.7 ) \times \mathbb I( \mathrm {Rank}\_{( 0.7, \mathbf s)} \leq 2)+ 0.7 \times \mathbb I (\mathrm {Rank}\_{(0.7, \mathbf s)} > 2) = 0.9 \times \mathbb I( 2 \leq 2)+ 0.7 \times \mathbb I (2 > 2) = 0.9$
>
>     $ \tilde s_{i3}^{\mathbf x^{\text {test}}}=( ( 0.9 - 0.3 ) + 0.3 ) \times \mathbb I( \mathrm {Rank}\_{( 0.3, \mathbf s)} \leq 2)+ 0.3 \times \mathbb I (\mathrm {Rank}\_{(0.3, \mathbf s)} > 2) = 0.9 \times \mathbb I( 3 \leq 2)+ 0.3 \times \mathbb I (3 > 2) = 0.3$
>
> 3.Algorithmic details.
> + Algorithm 1 in the manuscript describes the process of label binding. Likewise, taking $\mathbf s = [\mathbf {0.9}, \mathbf {0.7}, 0.3] \rightarrow \mathbf s^{'} = [\mathbf {0.9}, \mathbf {0.9}, 0.3]$ with **k** being **2** as an example,  since $\mathbf {0.9}$ and $\mathbf {0.7}$ are all within **top-2**, the logits of $\mathbf {0.9}$ and $\mathbf {0.7}$ are bound together $\rightarrow \mathbf {0.9}$ and $\mathbf {0.9}$. However, 0.3 is not in the **top-2**, so 0.3 will not be bound $\rightarrow 0.3$.
>
> [1]. TAI++:Text as Image for Multi-Label Image Classification by Co-Learning Transferable Prompt. IJCAI 2024

---

> ### Author Response · Authors · 2024-11-18
> **Official Response by Authors (2/2)**
>
> ### **Weakness 2: Discussion about the selection of baselines and underperformance reasons.**
> + Current mainstream Test-Time Adaptation (TTA) methods primarily adapt to **multi-class** instances by entropy minimization, with the core idea of increasing the prediction confidence of **top-1** label. However, for **multi-label** instances, focusing solely on the **top-1** label inevitably impairs the adaptation for other positive labels.
>
> + To our knowledge, our work is the first to investigate the feasibility of traditional entropy minimization in the multi-label setting. Therefore, we select the SOTA methods in the TTA area for multi-class scenarios as our baselines, including methods that do not require retaining historical knowledge (TPT [1], DiffTPT [2], RCLF [3]) and those that do (DMN [4], TDA [5]).
>
>     + For instance, DMN [4] introduces a dual-memory network that preserves historical knowledge from single-label instances, which intensifies the optimization bias towards the top-1 label when adapting to multi-label instances.
>
>     + TDA [5] proposes a dynamic key-value cache that retains only a small number of high-quality labels as key-value pairs at each step. Similar to DMN [4], it faces challenges in adapting to multi-label instances due to the erroneous accumulation of historical knowledge.
>
>     + DiffTPT [2] tends to neglect small object categories when generating multi-label pseudo-images, causing the model to focus more on optimizing prominent object categories.
>
>     + RLCF [3] employs teacher model logit distillation and more adaptation steps, which also results in excessive optimization for the top-1 label, thereby damaging the adaptation performance for other positive labels.
>
> [1]. Test-Time Prompt Tuning for Zero-Shot Generalization in Vision-Language Models. NeurIPS 2022
>
> [2]. Diverse Data Augmentation with Diffusions for Effective Test-time Prompt Tuning. ICCV 2023
>
> [3]. Test-Time Adaptation with CLIP Reward for Zero-Shot Generalization in Vision-Language Models. ICLR 2024
>
> [4]. Dual memory networks: A versatile adaptation approach for vision-language models. CVPR 2024
>
> [5]. Efficient Test-Time Adaptation of Vision-Language Models. CVPR 2024

---

> ### Author Response · Authors · 2024-11-23
> **Looking forward to your feedback.**
>
> Dear Reviewer, we are looking forward to your professional suggestions and hope to receive your guidance to further discuss and refine the contents of the work. We are eager for your response. Thank you.

---

> ### Author Response · Authors · 2024-11-25
> **Looking forward to your feedback**
>
> Dear Reviewer, your professional suggestions are crucial to our research. We kindly ask that you respond at your earliest convenience to further discuss and refine our work. We are eagerly awaiting your valuable feedback. Thank you!

---

> > ### Comment · Reviewer_97yB · 2024-11-26
> >
> > Thank you for your response; my concerns have been answered. I would like to keep my rating.

---

> > > ### Author Response · Authors · 2024-11-26
> > > **Response**
> > >
> > > Thanks for your time and response~

---

### Official Review · Reviewer_jmBy · 2024-11-03

**Soundness:** 2
**Presentation:** 2
**Contribution:** 3
**Rating:** 5
**Confidence:** 3

**Summary:**

This paper focuses on test time adaptation under a multi-label setting, this is an early work in this field.  This paper first analyzes why widely used entropy loss is not helpful in multi-label settings, and proposes a new method to adapt with multi-label. Then, the author proposes the view prompt and caption prompt to adapt the model for each instance. The experiments on three datasets show the effectiveness of the proposed method.

**Strengths:**

1. This paper focuses on an important question.
2. This paper has a good theoretical analysis.
3. The proposed method achieves better result than baselines.

**Weaknesses:**

1. The equ(6) is quite difficult to understand, more explanation is needed to show the meaning. The author should explain more about how weak labels and strong label is recognized in the proposed method, and the meaning of $\hat{s}_{ij}^{x^{test}$.
2. It is unclear which parameter is learnable in this method. The authors need to clearly point out all the learnable parameters.
3. The authors could explain more about the motivation of the view prompt and caption prompt, and why they are useful for this setting.

**Questions:**

1. The equ(6) is quite difficult to understand, more explanation is needed to show the meaning. The author should explain more about how weak labels and strong label is recognized in the proposed method, and the meaning of $\hat{s}_{ij}^{x^{test}$.
2. It is unclear which parameter is learnable in this method. The authors need to clearly point out all the learnable parameters.
3. The authors could explain more about the motivation of the view prompt and caption prompt, and why they are useful for this setting.

---

> ### Author Response · Authors · 2024-11-18
> **Official Response by Authors (1/2)**
>
> Thanks for your valuable suggestions, we will try to address your concerns and we are eager to engage in a more detailed discussion with you.
> ### **Weakness 1: Explanation of Eq.(6) and $\tilde s_{ij}^{\mathbf x^{\text {test}}}$. Recognize weak and strong label sets.**
> 1.Explanation of label binding and $\tilde s_{ij}^{\mathbf x^{\text {test}}}$ in Eq.(6) in manuscript.
> + Label binding refers to making the **top-k** predicted **logits** equal, as expressed below:
> $ \tilde s_{ij}^{\mathbf x^{\text {test}}}=( ( m_i^{\mathbf x^{\text {test}}} - s_{ij}^{\mathbf x^{\text {test}}} ) + s_{ij}^{\mathbf x^{\text {test}}} ) \times \mathbb I( \mathrm {Rank}\_{( s_{ij}^{\mathbf x^{\text {test}}}, \mathbf s_{i}^{\mathbf x^{\text {test}}})} \leq k^{\mathbf x_i^{\text {test}}} )+ s_{ij}^{\mathbf x^{\text {test}}} \times \mathbb I (\mathrm {Rank}\_{(s_{ij}^{\mathbf{x}^{\text {test}}}, \mathbf s_i^{\mathbf x^{\text {test}}})} > k^{\mathbf x_i^{\text {test}}} )$.
>
> + Since label binding (making ... equal) is non-differentiable, we employ the **stop-gradient** operation in VQ-VAE [1] for backpropagation, $i.e.$ $( ( m_i^{\mathbf x^{\text {test}}} - s_{ij}^{\mathbf x^{\text {test}}} ) + s_{ij}^{\mathbf x^{\text {test}}} )$ to perform label binding. Taking a **3-class** classification task as an example with class labels of **(1,2,3)**, assuming $k^{\mathbf x_i^{\text {test}}}$ is **2**, and the label binding process is $\mathbf s = [\mathbf {0.9}, \mathbf {0.7}, 0.3] \rightarrow \mathbf s^{'} = [\mathbf {0.9}, \mathbf {0.9}, 0.3]$. $\tilde s_{ij}^{\mathbf x^{\text test}}$ represents the **logit** of the **j-th** class in the **i-th** augmented view after label binding, $e.g.$, $\tilde s_{i2}^{\mathbf x^{\text test}}$ changes from $\mathbf {0.7}\rightarrow\mathbf{0.9}$. $m_i^{\mathbf x^{\text {test}}}$ denotes the maximum value of $\mathbf s$, which is $\mathbf {0.9}$. $\mathbb I(\cdot)$ is the indicator function. $\mathrm {Rank}\_{(a, \mathbf b)}$ indicates the descending rank of $a$ within **b**, $e.g.$, $\mathrm {Rank}\_{(0.7, \mathbf s)} = 2$. The process for computing the **bound logit** for each class is as follows:
>
>     $ \tilde s_{i1}^{\mathbf x^{\text {test}}}=( ( 0.9 - 0.9 ) + 0.9 ) \times \mathbb I( \mathrm {Rank}\_{( 0.9, \mathbf s)} \leq 2)+ 0.9 \times \mathbb I (\mathrm {Rank}\_{(0.9, \mathbf s)} > 2) = 0.9 \times \mathbb I( 1 \leq 2)+ 0.9 \times \mathbb I (1 > 2) = 0.9$
>
>     $ \tilde s_{i2}^{\mathbf x^{\text {test}}}=( ( 0.9 - 0.7 ) + 0.7 ) \times \mathbb I( \mathrm {Rank}\_{( 0.7, \mathbf s)} \leq 2)+ 0.7 \times \mathbb I (\mathrm {Rank}\_{(0.7, \mathbf s)} > 2) = 0.9 \times \mathbb I( 2 \leq 2)+ 0.7 \times \mathbb I (2 > 2) = 0.9$
>
>     $ \tilde s_{i3}^{\mathbf x^{\text {test}}}=( ( 0.9 - 0.3 ) + 0.3 ) \times \mathbb I( \mathrm {Rank}\_{( 0.3, \mathbf s)} \leq 2)+ 0.3 \times \mathbb I (\mathrm {Rank}\_{(0.3, \mathbf s)} > 2) = 0.9 \times \mathbb I( 3 \leq 2)+ 0.3 \times \mathbb I (3 > 2) = 0.3$
>
>     The logits after binding are $[\mathbf {0.9}, \mathbf {0.9}, 0.3]$, and we will introduce the process in detail in future version.
>
> 2.Recognize weak and strong label sets.
> + Given a test image $\mathbf x$, $\mathbf x$ is first augmented $N$ times to obtain different views {$\mathbf x_i|i=1,2,3,...,N$}. Then, for each $\mathbf x_i$, a most similar caption is retrieved to form $N$ view-caption pairs, defined as {$\langle \mathbf x_i, \mathbf t_i\rangle|i=1,2,3,...,N$}.
> + For example, given a pair $\langle \mathbf x_i, \mathbf t_i\rangle$, where $\mathbf t_i$ is **"A black bicycle parked in front of a car"**. We follow the nouns filter strategy in PVP [2] and extract label set **{bicycle, car}** from $\mathbf t_i$. This label set serves as the **strong** label set for $\mathbf t_i$ and also as the **weak** label set for $\mathbf x_i$. The term **"weak"** is called because $\mathbf t_i$ may not include all the labels presented in $\mathbf x_i$, for example, the truth label set of $\mathbf x_i$ could be **{bicycle, car, dog}**.
>
> [1]. Neural Discrete Representation Learning. NeurIPS 2017
>
> [2]. TAI++:Text as Image for Multi-Label Image Classification by Co-Learning Transferable Prompt. IJCAI 2024

---

> > ### Comment · Reviewer_jmBy · 2024-11-25
> > **Reply to authors**
> >
> > Thanks for your response. As you explain, I understand the weak label set is equal to the strong label set, Is my understanding right? If they are the same, what are their different effect in this method?

---

> ### Author Response · Authors · 2024-11-18
> **Official Response by Authors (2/2)**
>
> ### **Weakness 2: The learnable parameters.**
> + The learnable parameters in ML-TTA are the **view prompt** and **caption prompt** shown in Figure 2 in the manuscript. The image and text encoders of CLIP are frozen.
>
> ### **Weakness 3: Motivation of the view prompt and caption prompt. Why are they useful?**
> + The goal of ML-TTA is to enable adaptation to the multi-label test instance with varying distributions during the testing stage. Prompt tuning adapts to new data by adjusting the input context of the CLIP, thus not distorting the original knowledge of the pretrained CLIP model. Therefore, we also adopt prompt tuning strategy, treating prompt tuning at test-time as a way to furnish customized context for individual test instances.
>
> + Benefit from the aligned visual-language space of CLIP, the feature representations of images and texts share similar semantic information, therefore, the paired caption can be considered as **"pseudo image"** with accurate textual labels. This mitigates the potential limitation of weak label set, which may not fully capture the content of augmented views. Additionally, within the aligned space of CLIP, the model is to learn visual-related knowledge from text captions. Therefore, we adopt both view prompts and caption prompts to learn complementary information from views and captions jointly.

---

> ### Author Response · Authors · 2024-11-23
> **Looking forward to your feedback.**
>
> Dear Reviewer, we are looking forward to your professional suggestions and hope to receive your guidance to further discuss and refine the contents of the work. We are eager for your response. Thank you.

---

> ### Author Response · Authors · 2024-11-25
> **Reply to Reviewer jmBy**
>
> + We appreciate your feedback. Your understanding of the weak and strong label sets is right. They are the same in both **quantity and content** in ML-TTA. We differentiate them by their respective **action scopes**, hence the terms "weak" and "strong".
>
>     + **Weak Label Set**: Represents the **pseudo-true** labels for each augmented view. Since the true labels for each view are inaccessible and cannot be directly obtained. Therefore, we retrieve the most similar caption for each view and extract the textual labels to form the weak label set for that view. These textual labels, acting as an **approximation of the view's true labels**, provide as accurate label information as possible to the view.
>
>     + **Strong Label Set**: Represents the **known true labels** corresponding to each paired caption. Owing to the aligned visual-language space of CLIP, captions can be regarded as **pseudo-images with known true labels**. Therefore, the textual labels extracted from the caption are utilized directly as the true labels for the caption, which we refer to as the strong label set. These textual labels help the model capture visual-related knowledge from the caption and the aligned CLIP space.
>
> + Although the weak and strong label sets are the same in quantity and content, they differ in their **action scope**. The weak label set is defined by approximating the true labels of each view, whereas the strong label set is derived directly from the corresponding paired caption. In addition, we employ a confidence filtering strategy to filter out views and captions with high entropy (low confidence), ensuring that the label sets more accurately reflect the true label information of the views and captions.

---

> ### Author Response · Authors · 2024-11-30
> **Looking forward to your feedback**
>
> Dear reviewer jmBy, thanks for your previous suggestions for our work. We would like to further discuss the content with you and hope to receive your response to the manuscript.
>
> Additionally, if you find that the overall quality of the manuscript has improved after re-evaluating these modifications, we kindly ask you to consider adjusting the rating score accordingly.
>
> Looking forward to your feedback, thank you!

---

> ### Author Response · Authors · 2024-12-03
>
> Dear reviewer jmBy,
>
> Thanks again for your previous feedback. We wish to discuss the manuscript content with you and hope for your response.
>
> If you find the manuscript’s quality improved, we kindly request you to consider revising the rating score.
>
> Best regards,

---

### Author Response · Authors · 2024-11-22
**Thank you for considering our revisions and valuable sugestiongs. We are eager to engage in further discussions with you.**

Dear Reviewers, Area Chairs, Program Chairs, and Senior Area Chairs,

We address the reviewers' concerns with the following updates and improvements, and submit an improved manuscript highlighted in red:

1.    **Paired caption retrieval and label binding**: Detailed explanation of paired caption retrieval in Sec 3.3.1. Detailed explanation and example of label binding in  Sec 3.3.2 and Appendix B. Exploration to improve caption quality in Appendix C.

2.    **Learnable parameters.**:Add illustration of the learnable parameters in Figure 2.

3.    **View prompt and caption prompt**: Motivation and effect about view prompt and caption prompt in Sec 3.3.1.

4.    **Discussion about baselines**:Discussion on the selection of the baselines and analysis for their suboptimal performance in Sec 4.2.

5.    **Motivation about ML-TTA**:Clarified motivation about ML-TTA in Sec 1.

6.    **Complexity analysis**:Comparison experiment about adaptation complexity with TPT, DiffTPT, and RLCF in Sec 4.2.

7.    **Augmented view and evaluation metric**:Detailed explanation of augmented view in Sec 3.1, and mAP metric in Sec 4.1.

Thank you for considering our revisions and valuable suggestions. **We are grateful for your help with our work. If you have any further concerns, please do not hesitate to contact us and we look forward to discussing with you.**

---

### Meta-Review · Area_Chair_Ziuk · 2024-12-20

**Metareview:**

This paper introduces a novel technique, Bound Entropy Minimization (BEM), for multi-label test-time adaptation (ML-TTA). Unlike existing methods that prioritize the most confident prediction, BEM enhances the confidence of the top-k predicted labels simultaneously, effectively addressing the challenges of ML-TTA. The paper presents comprehensive experimental evaluations across several datasets, including MSCOCO, VOC, and NUSWIDE, demonstrating that the ML-TTA framework with BEM outperforms current state-of-the-art methods. The structure is clear, and both the methodology and results are well-presented. Although the initial submission lacked some clarity in the algorithm description and experimental interpretation, the authors have successfully addressed these concerns in the rebuttal, leading to a significant improvement in the overall presentation. Therefore, I recommend accepting this paper.

**Additional Comments On Reviewer Discussion:**

After the rebuttal, the major reviewers' concerns were addressed (except for one reviewer who did not provide further response), and one reviewer increased their score accordingly.

---

### Decision · Program_Chairs · 2025-01-22

Accept (Poster)